# ROBUST DOMAIN GENERALIZATION UNDER DIVERGENT MARGINAL AND CONDITIONAL DISTRIBUTIONS

## ABSTRACT

Domain generalization (DG) aims to learn predictive models that can generalize to unseen domains. Most existing DG approaches focus on learning domain-invariant representations under the assumption of conditional distribution shift (i.e., they primarily address changes in $P(X|Y)$ while assuming the label marginal $P(Y)$ remains stable). However, real-world data seldom satisfy this assumption. Multiple domains often differ in more complex ways, where both the label distribution $P(Y)$ and the conditional distribution $P(X|Y)$ vary simultaneously. In this work, we propose a new framework for robust domain generalization under divergent marginal and conditional distributions. We introduce a novel risk bound for unseen domains by explicitly decomposing the joint distribution into marginal and conditional components and characterizing risk gaps arising from both sources of divergence. To operationalize this bound, we design a meta-learning procedure that minimizes and validates the proposed risk bound across seen domains, ensuring strong generalization to unseen ones. Empirical evaluations demonstrate that our method achieves state-of-the-art performance not only on conventional DG benchmarks but also in challenging Multi-Domain Long-Tailed Recognition (MDLT) settings where both marginal and conditional shifts are pronounced.

## 1 INTRODUCTION

Modern machine learning models are predominantly trained on static, homogeneous datasets where the joint distribution over inputs $X$ and labels $Y$ is assumed to remain consistent. However, real-world deployments rarely afford such stability. Once models move beyond their curated training domains and begin operating in real-world environments, they must contend with shifts in geography, time, sensors, users, or even tasks. These changes often introduce compound distributional shifts that severely degrade performance.

Two major axes of distributional shift commonly co-occur in practice: shifts in the marginal label distribution $P(Y)$ and shifts in the class-conditional distribution $P(X \mid Y)$. The former reflects changes in class frequencies, which naturally arise across populations, seasons, or use cases. For instance, a species that is rare in one habitat may be abundant in another. The latter encompasses changes in the appearance or representation of each class, due to variations in context, style, modality, or sensor characteristics. For example, the concept of a "cat" might appear as a sketch in one domain and a thermal image in another.

Despite their practical relevance, most prior work in domain generalization (DG) assumes that only the conditional distribution $P(X \mid Y)$ varies across domains, while $P(Y)$ remains fixed (Zhou et al., 2022). This assumption underpins many popular DG methods that focus on learning domain-invariant representations. However, these approaches neglect the impact of marginal shifts, which can significantly affect model calibration and decision thresholds, especially in long-tailed or imbalanced settings (Xia et al., 2023). In such cases, models may become overconfident in dominant classes, under-represent rare classes, and fail to adjust to shifting class priors, leading to degraded performance in unseen domains (Tan et al., 2024). On the other hand, approaches aimed at addressing label imbalance or long-tail distributions often operate within a single domain, failing to account for representational drift across environments (Fu & Yusof, 2024). Thus, there remains a critical gap:

*how can we build models that generalize robustly under simultaneous marginal and conditional distribution shifts?*

In this work, we take a principled approach to tackle this challenge. We begin by revisiting the upper bounds for domain generalization, explicitly decomposing the joint distribution shift into marginal and conditional components. Our theoretical analysis reveals distinct risk contributions from each axis of divergence and motivates a new risk bound tailored for unseen domains. Building on this insight, we propose a novel meta-learning framework that minimizes and validates this bound across a diverse set of source domains. Our method trains models to actively anticipate and counteract shifts in both $P(Y)$ and $P(X \mid Y)$, rather than passively hoping for invariant features.

To evaluate our approach, we conduct extensive experiments on standard domain generalization benchmarks as well as designed multi-domain long-tailed recognition scenarios where both sources of shift are pronounced. Our results demonstrate that the proposed method not only outperforms state-of-the-art DG methods but also exhibits remarkable robustness in the face of compounded distributional shifts.

In summary, this paper makes the following key contributions:

- We derive a new theoretical upper bound of the risk in the query domain by the risk in the mixture of the support domains, which accounts for the divergence in both marginal and conditional distributions.

- Inspired by theoretical bound, we develop a meta-learning based domain generalization algorithm called RC-Align, which takes into account both domain and label shifts.

- We empirically show that RC-Align outperforms existing methods on both the standard DG and challenging Multi-Domain Long-Tailed Recognition (MDLT) benchmarks.

## 2 RELATED WORK

**Domain Generalization (DG) under divergent distributions.** Classical domain generalization (DG) aims to learn models that perform well on unseen domains by mitigating shifts in the class-conditional distribution $P(X \mid Y)$. Approaches include domain-invariant representation learning such as correlation alignment (Sun & Saenko, 2016), adversarial feature alignment (Ganin et al., 2016), and conditional invariance (Li et al., 2018d), as well as optimization-based techniques (Wang et al., 2023; Cha et al., 2021) and augmentation strategies (Zhou et al., 2021; Verma et al., 2019). However, most DG methods implicitly assume balanced label distributions across domains.

In contrast, long-tailed (LT) recognition focuses on shifts in the marginal label distribution $P(Y)$, where a few head classes dominate while many tail classes are severely under-represented. Remedies include resampling (Buda et al., 2018), reweighting (Wang et al., 2017; Lin et al., 2017), logit adjustment (Ren et al., 2020), and two-stage or ensemble strategies (Kang et al., 2020; Wang et al., 2021; Iscen et al., 2021). These have proven effective in single-domain LT settings but struggle under domain shifts.

The compounded challenge defines *multi-domain long-tailed recognition (MDLT)*, where both $P(X \mid Y)$ and $P(Y)$ diverge across domains. Tail-class representations become especially fragile under such joint shifts (Gu et al., 2022). Recent works attempt to address this by explicitly defining domain–class transferability (Yang et al., 2022), modeling imbalanced domain adaptation (Ding et al., 2023), or tackling long-tailed recognition under domain shifts with a meta-learning based framework (Gu et al., 2022). A key insight is that label imbalance is an intrinsic but overlooked aspect of DG: beyond MDLT, addressing cross-domain imbalance improves standard DG benchmarks and complements existing DG algorithms, suggesting that label imbalance is largely orthogonal to prior DG remedies (Yang et al., 2022). Despite progress, a unified framework that jointly tackles conditional shifts and marginal imbalance remains an open direction for robust DG.

**Meta-Learning for DG.** Meta-learning has been widely applied to domain generalization by simulating domain shifts in episodic training. Early bilevel approaches such as MLDG (Li et al., 2018a) and MetaReg (Balaji et al., 2018) encourage robustness by splitting source domains or learning transferable regularizers. Later works emphasize gradient-based objectives, including update alignment (Shi et al., 2021) and gradient averaging (Wang et al., 2025), while others target invariant representation learning through bilevel consistency (Jia & Zhang, 2024). Distinct from these methods, our algorithm

introduces a *robust compound alignment* framework that integrates manifold mixup with episodic adaptation, jointly addressing cross-domain discrepancy and intra-domain imbalance beyond prior gradient- or representation-focused approaches.

**Risk Bounds and Theoretical Foundations under Distribution Shifts.** An important line of research provides theoretical guarantees for learning under distribution shifts. Yang & Xu (2020) investigate long-tailed regimes, demonstrating how semi- and self-supervised signals can alleviate bias and improve generalization under severe imbalance. Extending this perspective, Yang et al. (2022) formalize domain–class relations in multi-domain long-tailed recognition and derive a bound that bounds transferability statistics, that is highly correlated with test accuracy. More recently, Yang et al. (2025) focus on robust domain adaptation, introducing a margin-based divergence and establishing generalization bounds on adversarially robust target risk. Building upon these advances, our work develops a unified risk bound that explicitly decomposes both marginal and conditional divergences, and further embeds this insight into a meta-learning framework—advancing the theoretical foundations while translating them into practical robustness under compounded distribution shifts.

## 3  UPPER BOUND FOR DOMAIN GENERALIZATION UNDER COMPOUND SHIFTS

A central challenge in domain generalization (DG) is evaluating how well a model trained on a collection of source domains will perform on an unseen target domain. Since no data from the target domain is available during training, a natural and widely adopted framework is the leave-one-domain-out (LODO) setting, where one domain is held out as a proxy for the unseen target, and the model is trained on the remaining source domains. This episodic formulation not only mimics real-world deployment conditions but also provides a rigorous setting for analyzing domain shift sensitivity and generalization behavior.

Formally, we consider $K$ source domains $\mathbb{D} = \{\mathcal{D}_1, \ldots, \mathcal{D}_K\}$. The model consists of a feature extractor $f_\theta$ and a classifier $g_\phi$, with combined parameters $\Theta = \{\theta, \phi\}$. In each LODO episode, we designate one domain $\mathcal{D}_i$ as the query domain (unseen during training) and the remaining domains $\mathcal{D}_{-i}$ as the support mixture. The goal is to understand and control the unseen-domain risk $\mathcal{R}_{\mathcal{D}_i}(\Theta)$, defined as:

$$\mathcal{R}_{\mathcal{D}_i}(\Theta) = \mathbb{E}_{(x,y) \sim \mathcal{D}_i} \left[ \ell \left( g_\phi \left( f_\theta \left( x \right) \right), y \right) \right],$$

where $\ell$ is a bounded, $L_\ell$-Lipschitz loss function with respect to the distance metric d in the feature space $\mathcal{Z}$ (See Appendix C.1 for details).

However, measuring the absolute value of $\mathcal{R}_{\mathcal{D}_i}(\Theta)$ provides limited insight into what causes generalization failures. To make this analysis actionable, we consider how and why the query (i.e., unseen, target) domain risk deviates from the risk over the support mixture (i.e., source) domain $\mathcal{R}_{\mathcal{D}_{-i}}(\Theta)$ (i.e. $\mathcal{R}_{-i}(\Theta) = \sum_{k \neq i} P(\mathcal{D} = \mathcal{D}_k) \mathcal{R}_k(\Theta)$). This motivates a decomposition of the generalization gap $\mathcal{R}_{\mathcal{D}_i}(\Theta) - \mathcal{R}_{\mathcal{D}_{-i}}(\Theta)$.

To achieve a tractable and interpretable analysis, we propose to decompose each domain's joint distribution into two components: (1) the marginal label distribution (i.e., prior) $P(Y = c|\mathcal{D}_i) := \pi_i(c)$, and (2) the class-conditional feature distribution $P(Z = f_\theta(x) \mid Y = c, \mathcal{D}_i) := P_{i,c}$.

This decomposition enables us to separate the sources of domain shift into two orthogonal axes: (1) Prior mismatch: how the label frequencies differ across domains, and (2) Feature mismatch: how the conditional feature distributions shift for each class.

Such decomposition is crucial because existing DG methods often treat domain shift as a single nuisance variable (e.g., via domain-invariant representations), without disentangling what kind of shift is occurring. In contrast, our analysis shows that even if features are invariant, shifts in label priors alone can lead to significant performance degradation and vice versa. This leads to our first key theoretical result:

**Assumption 1.** *The loss function $\ell$ is bounded and $L_\ell$-Lipschitz continuous with respect to the distance metric d in the feature space $\mathcal{Z}$.*

**Theorem 1.** *[Query–support bound] Under* **??** *1, The risk on a query domain ($\mathcal{D}_i$) is bounded by the risk on the support mixture ($\mathcal{D}_{-i}$) plus mismatch terms:*

$$\mathcal{R}_{\mathcal{D}_i}(\Theta) \leq \mathcal{R}_{\mathcal{D}_{-i}}(\Theta) + \underbrace{\sum_{c \in \mathcal{C}} |\pi_i(c) - \pi_{-i}(c)| \cdot \mathcal{R}_{-i,c}(\Theta)}_{\text{Prior Shift}} + \underbrace{L_\ell \sum_{c \in \mathcal{C}} \pi_i(c) \, \mathsf{W}_1\big(P_{i,c}, P_{-i,c}\big)}_{\text{Feature Shift}}, \quad (1)$$

*where $\mathsf{W}_1(\cdot, \cdot)$ $L_\ell$ are 1-Wasserstein distance and a lipschitz constant of $\ell$, respectively.*

All proofs of theoretical results in this section (Theorems 1 to 3, lemma 1, and proposition 1) are provided in Appendices A and B.

**Implications of Theorem 1.** This theorem explicitly separates the challenges posed by compound shifts. Crucially, the impact of the Prior Shift is modulated by the actual class-wise support risks, $\mathcal{R}_{-i,c}(\Theta)$, rather than a constant bound. Minimizing the overall support risk $\mathcal{R}_{\mathcal{D}_{-i}}(\Theta)$ (via Cross-Entropy) inherently minimizes $\mathcal{R}_{-i,c}(\Theta)$, thereby actively controlling the adverse effects of label distribution mismatch.

### 3.1 Controlling the Query-Support Gap: A Tractable Bound

While Theorem 1 identifies the sources of the gap for domain generalization ($\mathcal{R}_{\mathcal{D}_i}(\Theta) - \mathcal{R}_{\mathcal{D}_{-i}}(\Theta)$), the Feature Shift term involves the intractable 1-Wasserstein distance ($\mathsf{W}_1$) between feature distributions. We now establish a connection between this term and the practical Domain-Class Distribution Alignment (DA) loss (Yang et al., 2022) ($\mathcal{L}_{\text{DA}}$) used in our algorithm.

We utilize a series of analytical steps involving an InfoNCE decomposition of the DA loss and a two-step transport argument that routes mass through the source centroids. This allows us to bound the Wasserstein distance using the expected DA loss. (Detailed derivations involving Lemma 2, Lemma 3, and Lemma 4 are provided in Appendix C.2).

**Theorem 2.** *[DA loss controls the query–support gap] The total feature distribution mismatch (Feature Shift term) is upper-bounded by the DA loss:*

$$\sum_c \pi_i(c) \, \mathsf{W}_1\big(P_{i,c}, P_{-i,c}\big) \leq \mathbb{E} \, \mathcal{L}_{\text{DA}}^{(i)}(\theta) + C_{\text{scat}}^{(i)} + R, \quad (2)$$

*where $C_{\text{scat}}^{(i)}$ is the average weighted scatter within the support domains (defined formally in Appendix C.2), and $R$ is the diameter of the feature space.*

**The Final Theoretical Bound.** Combining Theorem 1 and Theorem 2 yields our main theoretical result, connecting the unobserved target risk to concrete, optimizable quantities:

$$\mathcal{R}_{\mathcal{D}_i}(\Theta) \leq \mathcal{R}_{\mathcal{D}_{-i}}(\Theta) + \sum_c |\Delta\pi_c| \mathcal{R}_{-i,c}(\Theta) + L_\ell \Big( \mathbb{E} \, \mathcal{L}_{\text{DA}}^{(i)}(\theta) + C_{\text{scat}}^{(i)} + R \Big), \quad (3)$$

where $\Delta\pi_c = \pi_i(c) - \pi_{-i}(c)$.

**Significance of the Bound.** This bound (Eq. (3)) provides the justification for our proposed methodology:

- **Controlling Feature Shift:** Our analysis proves that the practical DA loss $\mathcal{L}_{\text{DA}}$ directly upper-bounds the feature mismatch term. This provides a stronger, risk-based justification than the statistical bounds in prior work (Yang et al., 2022).

- **Controlling Prior Shift:** The impact of prior mismatch is modulated by the model's class-wise source performance $\mathcal{R}_{-i,c}(\Theta)$.

- **Synergistic Optimization:** Our composite objective minimizes this bound through complementary mechanisms. $\mathcal{L}_{\text{CE}}$ primarily reduces the source risks $\mathcal{R}_{-i,c}(\Theta)$. Crucially, optimizing for classification accuracy inherently promotes intra-class compactness in the feature space, thereby *implicitly controlling* the scatter term $C_{\text{scat}}^{(i)}$ (which measures the average intra-domain spread). Concurrently, $\mathcal{L}_{\text{DA}}$ explicitly minimizes the inter-domain feature mismatch, ensuring that these compact class clusters are aligned across domains.

## 3.2 Generalization to Arbitrary Unseen Target Domains

We generalize the analysis to an arbitrary unseen target domain $\mathcal{T}$ by relating it to a mixture of source domains, $\widetilde{\mathcal{T}}_{\boldsymbol{\pi}} := \sum_{i=1}^{K} \pi_i \mathcal{D}_i$, where $\boldsymbol{\pi} \in \Delta^K$ (the $K$-dimensional probability simplex).

**Theorem 3.** *[Mixture-aware target bound] For any model $\Theta$ and $\boldsymbol{\pi} \in \Delta^K$, the risk on $\mathcal{T}$ is bounded by:*

$$\mathcal{R}_{\mathcal{T}}(\Theta) \ \leq \ \sum_{i=1}^{K} \pi_i \, \mathcal{R}_{\mathcal{D}_i}(\Theta) \ + \ L_\ell \, \mathsf{W}_1\big(\mathcal{T}, \widetilde{\mathcal{T}}_{\boldsymbol{\pi}}\big) \, . \tag{4}$$

Combining Theorem 3 with the final theoretical bound (Eq. (3)) yields the comprehensive guarantee (detailed in Appendix C.6), explicitly showing how controlling the terms leads to robust generalization on arbitrary target domains.

## 3.3 Analysis of the Meta-Learning Procedure

We analyze the effectiveness of the meta-learning framework in minimizing the derived bound. We define the population inner objective $J_{-i}(\Theta)$ (combining risk and alignment) and the adaptation step $\Theta_i^+ := \Theta - \alpha \nabla_\Theta J_{-i}(\Theta)$.

To gain insights into the optimization dynamics of the inner loop, we analyze the convergence behavior under the standard assumptions of $L$-smoothness and the Polyak–Lojasiewicz (PL) condition (**?? 2** in Appendix C.7). While the PL condition is a strong assumption for the non-convex landscapes of deep networks, it serves as a standard analytical tool (Karimi et al., 2016) to characterize local convergence behavior and understand the adaptation dynamics (See discussion in Appendix C.7).

**Lemma 1.** *[One-Step Contraction] Under standard smoothness/PL conditions, the inner adaptation step yields a geometric contraction of the objective:*

$$J_{-i}(\Theta_i^+) - J_{-i}^\star \leq \rho(\alpha)\big(J_{-i}(\Theta) - J_{-i}^\star\big), \tag{5}$$

*where the contraction factor $\rho(\alpha) < 1$.*

This contraction suggests that the inner loop makes consistent progress within this idealized setting. We further analyze how this impacts the DA alignment term specifically.

**Proposition 1.** *[Reduction of the DA alignment term] The DA loss on the source domains after adaptation ($\Theta_i^+$) is bounded as:*

$$\mathbb{E}\mathcal{L}_{\mathrm{DA}}^{(-i)}(\Theta_i^+) \leq \frac{\rho(\alpha)}{\lambda_{\mathrm{DA}}} J_{-i}(\Theta) + \frac{1 - \rho(\alpha)}{\lambda_{\mathrm{DA}}} J_{-i}^\star. \tag{6}$$

Proposition 1 confirms that the inner adaptation step actively reduces the alignment objective at a geometric rate. The MLDG (Li et al., 2018a)-style outer loop ensures that this improved source alignment translates to better target generalization by minimizing the bound established in Eq. (3). A further discussion on optimization dynamics is provided in Appendix C.7.

## 4 Proposed Methodology: Robust Compound Alignment

Based on the theoretical insights from Section 3, we propose **RC-Align** (Robust Compound Alignment), a meta-learning framework designed to minimize the upper bound for domain generalization (Eq. (3)) under simultaneous marginal and conditional distribution shifts.

### 4.1 Loss Functions: Controlling Prior and Feature Shifts

To operationalize the bound in Eq. (3), we use a composite objective function that addresses both the Prior Shift Impact and the Feature Shift terms.

**Classification Loss (Controlling Prior Shift Impact).** Theorem 1 shows that the impact of prior shift is modulated by the class-wise support risks $\mathcal{R}_{-i,c}(\Theta)$. We use the standard empirical Cross-Entropy

(CE) loss to minimize these risks. Given a batch $B$:

$$\widehat{\mathcal{L}}_{\text{CE}}(B; \Theta) = \frac{1}{|B|} \sum_{(x,y) \in B} \text{CE}(g_\phi(f_\theta(x)), y). \tag{7}$$

By improving classification accuracy on the source domains, $\widehat{\mathcal{L}}_{\text{CE}}$ actively reduces the sensitivity of the model to changes in label priors.

**Domain-Class Distribution Alignment (DA) Loss (Controlling Feature Shift).** Theorem 2 proves that the DA loss upper-bounds the feature mismatch term $\text{W}_1(P_{i,c}, P_{-i,c})$. We employ a contrastive loss to pull features closer to the centroids of the same class in other domains, thereby minimizing this mismatch. Given a batch $B$ and pre-computed support centroids $\{\mu\}$:

$$\widehat{\mathcal{L}}_{\text{DA}}(B; \theta, \{\mu\}) = \frac{1}{|B|} \sum_{(x,y,i) \in B} \frac{-1}{K-1} \sum_{j \neq i} \log \frac{\exp\left(-\text{d}(f_\theta(x), \mu_{j,y})\right)}{\sum_{(j',c') \neq (i,y)} \exp\left(-\text{d}(f_\theta(x), \mu_{j',c'})\right)}. \tag{8}$$

**Total Training Objective.** The objective used in the inner loop is the empirical weighted combination, directly targeting the optimizable terms of the upper bound for domain generalization:

$$\widehat{\mathcal{L}}_{\text{total}}(B; \Theta, \{\mu\}) = \widehat{\mathcal{L}}_{\text{CE}}(B; \Theta) + \lambda_{\text{DA}} \widehat{\mathcal{L}}_{\text{DA}}(B; \theta, \{\mu\}), \tag{9}$$

where $\lambda_{\text{DA}}$ is a regularization parameter that balances the two terms.

### 4.2 THE RC-ALIGN WITH A META-LEARNING FRAMEWORK

Our proposed method, **RC-Align**, utilizes a two-stage, MAML (Finn et al., 2017)-style procedure within the Leave-One-Domain-Out (LODO) protocol to ensure that the minimization of the composite objective translates to robust generalization on the unseen domain. The framework is detailed in Algorithm 1.

**Inner Adaptation Loop.** In the inner loop, for a given held-out virtual target domain $\mathcal{D}_i$, the model's meta-parameters $\Theta$ are temporarily adapted using the remaining source domains $\mathcal{D}_{-i}$. This adaptation is guided by our composite objective (Eq. (9)), which requires pre-computed class centroids $\{\mu\}_{-i}$ from the source domains. For computational efficiency within the MAML framework, we estimate the centroids $\{\mu\}_{-i}$ on-the-fly using the features from the current mini-batches of the support domains (Line 6 in Algorithm 1). While batch-based estimation can introduce noise, particularly for tail classes in the MDLT setting, the meta-learning process inherently learns to adapt robustly despite this noise. Furthermore, the regularization provided by Manifold Mixup (Verma et al., 2019) helps stabilize the optimization. (A further discussion is provided in Appendix C.7.) This aims to minimize both source risk and feature misalignment simultaneously.

$$\Theta' \leftarrow \Theta - \alpha \nabla_\Theta \widehat{\mathcal{L}}_{\text{total}}(\Theta; B_{-i}, \{\mu\}_{-i}). \tag{10}$$

As proven in Section 3.3 (Proposition 1), this step guarantees a reduction in the alignment objective.

**Outer Meta-Optimization Loop.** In the outer loop, we evaluate the performance of the adapted model $\Theta'$ on the previously held-out virtual target domain $\mathcal{D}_i$. The resulting meta-loss provides a gradient signal to update the original meta-parameters $\Theta$:

$$\Theta \leftarrow \Theta - \eta \nabla_\Theta \widehat{\mathcal{L}}_{\text{CE}}(B_i; \Theta').$$

This meta-optimization process trains the model to find an initial set of parameters $\Theta$ that is primed for rapid and effective adaptation, ensuring that the reduction in the theoretical bound (achieved in the inner loop) results in improved performance on the unseen target domain. We use the efficient first-order MAML (FO-MAML) for this step. For detailed algorithm, refer to Appendix D.

## 5 EXPERIMENTS

We evaluate RC-Align on two challenging settings: (1) standard Domain Generalization (DG) and (2) Multi-Domain Long-Tailed Recognition (MDLT).

## 5.1 Experimental Setup

**Datasets.** We evaluate RC-Align across two distinct scenarios. For standard Domain Generalization (DG), we utilize four widely adopted benchmarks: PACS (Li et al., 2017), VLCS (Fang et al., 2013), OfficeHome (Venkateswara et al., 2017), and TerraIncognita (Beery et al., 2018). To assess performance under challenging compound shifts (where both $P(Y)$ and $P(X|Y)$ diverge), we employ five Multi-Domain Long-Tailed Recognition (MDLT) benchmarks. Following established protocols (Yang et al., 2022), these include the long-tailed versions of the DG datasets (PACS-MLT, VLCS-MLT, OfficeHome-MLT, TerraIncognita-MLT) and the large-scale DomainNet-MLT (Peng et al., 2019). These MDLT setups induce significant and divergent long-tail class distributions across domains.

**Metrics.** We follow the standard Leave-One-Domain-Out (LODO) evaluation protocol. We report the average accuracy across all held-out target domains (Average). For the MDLT setting, where robustness is critical, we additionally report the accuracy on the worst-performing domain (Worst) to assess the model's resilience under severe compound shifts.

**Implementation Details.** We implement RC-Align within the DomainBed framework (Gulrajani & Lopez-Paz, 2021) to ensure reproducibility and fair comparison. Following standard protocol, we utilize a ResNet-50 (He et al., 2016) backbone pretrained on ImageNet (Deng et al., 2009). Models are optimized using the Adam optimizer (Kingma & Ba, 2015). We follow the training-domain validation procedure for hyperparameter selection. All experiments were conducted using NVIDIA RTX 3090 GPUs. Comprehensive implementation details, including dataset statistics, training protocols, and full hyperparameter configurations, are provided in Appendix E.

## 5.2 Standard Domain Generalization (DG) Results

We compare our method against established DG algorithms. For more detailed results, refer to Appendix G.

Table 1: Summary of average domain generalization accuracy (%) across four datasets.

| Baselines (Part 1) | | | | | | Baselines (Part 2) | | | | | |
|---|---|---|---|---|---|---|---|---|---|---|---|
| **Algorithm** | **PACS** | **VLCS** | **OfficeHome** | **TerraInc.** | **Avg.** | **Algorithm** | **PACS** | **VLCS** | **OfficeHome** | **TerraInc.** | **Avg.** |
| ERM | 85.5 | 77.5 | 66.5 | 46.1 | 68.9 | ARM | 85.1 | 77.6 | 64.8 | 45.5 | 68.3 |
| IRM | 83.5 | 78.5 | 64.3 | 47.6 | 68.5 | VREx | 84.9 | 78.3 | 66.4 | 46.4 | 69.0 |
| GroupDRO | 84.4 | 76.7 | 66.0 | 43.2 | 67.6 | RSC | 85.2 | 77.1 | 65.5 | 46.6 | 68.6 |
| Mixup | 84.6 | 77.4 | 68.1 | 47.9 | 69.5 | MetaReg | 83.6 | 76.7 | 67.6 | 48.2 | 69.0 |
| MLDG | 84.9 | 77.2 | 66.8 | 47.7 | 69.2 | MLIR | 86.8 | 80.7 | 69.8 | 51.0 | 72.1 |
| CORAL | 86.2 | 78.8 | 68.7 | 47.6 | 70.3 | Mixstyle | 85.2 | 77.9 | 60.4 | 44.0 | 66.9 |
| MMD | 84.6 | 77.5 | 66.3 | 42.2 | 67.7 | SWAD | 88.1 | 79.1 | 70.6 | 50.0 | 72.0 |
| DANN | 83.7 | 78.6 | 65.9 | 46.7 | 68.7 | PCL | 88.7 | 78.0 | 71.6 | 52.1 | 72.6 |
| CDANN | 82.6 | 77.5 | 65.8 | 45.8 | 67.9 | BoDA | 86.9 | 78.5 | 69.3 | 50.2 | 71.2 |
| MTL | 84.6 | 77.2 | 66.4 | 45.6 | 68.5 | SAGM | 86.6 | 80.0 | 70.1 | 48.8 | 71.4 |
| SagNet | 86.3 | 77.8 | 68.1 | 48.6 | 70.2 | iDAG | **88.8** | 76.9 | **71.8** | 46.1 | 70.9 |
| GMDG | 85.6 | 79.2 | 70.7 | 50.1 | 71.4 | Arith | 86.5 | 79.4 | 69.4 | 48.1 | 70.9 |
| **Ours** | 87.5 | **81.0** | 70.9 | **52.6** | **73.0** | **Ours** | 87.5 | **81.0** | 70.9 | **52.6** | **73.0** |

**Analysis.** As shown in Table 1, RC-Align achieves state-of-the-art performance, securing the highest average accuracy across all four standard DG benchmarks. Notably, our method sets a new top score on VLCS, and TerraIncognita and demonstrates highly competitive results on PACS, and OfficeHome, consistently ranking among the top-performing methods. The substantial 3.8% average accuracy improvement over MLDG, another meta-learning baseline, is particularly telling. This highlights the effectiveness of our core proposal: incorporating an explicit Domain-Class Distribution Alignment (DA) loss within the meta-adaptation phase. While MLDG learns a generalizable initialization, our approach ensures this initialization is also primed to align class-conditional feature distributions, leading to more robust generalization even when class priors are balanced.

## 5.3 Multi-Domain Long-Tailed Recognition (MDLT) Results

We evaluate the robustness of RC-Align under compound shifts, comparing against specialized LT/MDLT methods. In all MDLT experiments, baseline results are from Yang et al. (2022). We defer detailed descriptions of each baseline to Appendix F, and detailed results in Appendix H.

Table 2: Consolidated Mean (Average) and Worst Domain Accuracy across MDLT Benchmarks.

| Algorithm | VLCS-MLT | | PACS-MLT | | OfficeHome-MLT | | TerraInc-MLT | | DomainNet-MLT | |
|---|---|---|---|---|---|---|---|---|---|---|
| | Average | Worst | Average | Worst | Average | Worst | Average | Worst | Average | Worst |
| ERM | 76.3 ±0.4 | 53.6 ±1.1 | 97.1 ±0.1 | 95.8 ±0.2 | 80.7 ±0.0 | 71.3 ±0.1 | 75.3 ±0.3 | 67.4 ±0.3 | 58.6 ±0.2 | 29.4 ±0.3 |
| IRM | 76.5 ±0.2 | 52.3 ±0.7 | 96.7 ±0.2 | 95.2 ±0.4 | 80.6 ±0.4 | 70.7 ±0.2 | 73.3 ±0.7 | 64.3 ±1.3 | 57.1 ±0.1 | 27.6 ±0.1 |
| GroupDRO | 76.7 ±0.4 | 54.1 ±1.3 | 97.0 ±0.1 | 95.3 ±0.4 | 80.1 ±0.3 | 68.7 ±0.9 | 72.0 ±0.4 | 66.6 ±0.2 | 53.6 ±0.1 | 25.9 ±0.3 |
| Mixup | 75.9 ±0.1 | 52.7 ±1.3 | 96.7 ±0.2 | 95.1 ±0.2 | 81.2 ±0.2 | 72.3 ±0.6 | 71.1 ±0.7 | 60.4 ±1.1 | 57.6 ±0.1 | 28.7 ±1.0 |
| MLDG | 76.9 ±0.2 | 53.6 ±0.5 | 96.6 ±0.1 | 94.1 ±0.3 | 80.4 ±0.2 | 70.2 ±0.6 | 76.6 ±0.2 | 66.9 ±0.5 | 58.5 ±0.0 | 28.7 ±0.0 |
| CORAL | 75.9 ±0.5 | 51.6 ±0.7 | 96.6 ±0.5 | 94.3 ±0.7 | 81.9 ±0.1 | 72.7 ±0.6 | 76.4 ±0.5 | 67.8 ±0.9 | 59.4 ±0.1 | 30.1 ±0.4 |
| MMD | 76.3 ±0.6 | 53.4 ±0.3 | 96.9 ±0.1 | 96.2 ±0.2 | 78.4 ±0.4 | 70.7 ±0.8 | 73.3 ±0.4 | 63.7 ±1.1 | 56.7 ±0.0 | 27.2 ±0.2 |
| DANN | 77.5 ±0.1 | 54.1 ±0.3 | 96.5 ±0.0 | 94.3 ±0.1 | 79.2 ±0.2 | 67.7 ±0.9 | 68.7 ±0.9 | 61.1 ±1.0 | 55.8 ±0.1 | 26.9 ±0.4 |
| CDANN | 76.6 ±0.4 | 53.6 ±0.4 | 96.1 ±0.1 | 94.5 ±0.3 | 79.0 ±0.2 | 69.4 ±0.3 | 70.3 ±0.5 | 65.9 ±1.0 | 56.0 ±0.1 | 27.7 ±0.4 |
| MTL | 76.3 ±0.3 | 52.9 ±0.5 | 96.7 ±0.2 | 94.5 ±0.6 | 79.5 ±0.2 | 69.8 ±1.6 | 75.0 ±0.7 | 67.7 ±1.4 | 58.6 ±0.1 | 29.3 ±0.2 |
| SagNet | 76.3 ±0.2 | 52.3 ±0.2 | 97.2 ±0.1 | 95.2 ±0.3 | 80.9 ±0.1 | 70.5 ±0.5 | 75.1 ±1.6 | 66.5 ±2.1 | 58.9 ±0.0 | 29.4 ±0.2 |
| Fish | 77.5 ±0.3 | 54.3 ±0.4 | 96.9 ±0.2 | 95.2 ±0.2 | 81.3 ±0.3 | 71.3 ±0.5 | 75.3 ±0.5 | 66.3 ±0.5 | 59.6 ±0.1 | 29.1 ±0.1 |
| Focal | 75.6 ±0.4 | 52.3 ±0.2 | 96.5 ±0.2 | 94.6 ±0.7 | 77.9 ±0.0 | 67.6 ±0.4 | 75.7 ±0.4 | 65.3 ±1.1 | 57.8 ±0.2 | 27.5 ±0.1 |
| CBLoss | 76.8 ±0.3 | 52.5 ±0.5 | 96.9 ±0.1 | 95.1 ±0.4 | 79.8 ±0.2 | 69.5 ±0.7 | 78.0 ±0.4 | 68.3 ±2.0 | 58.9 ±0.1 | 30.1 ±0.3 |
| LDAM | 77.5 ±0.1 | 52.9 ±0.2 | 96.5 ±0.2 | 94.7 ±0.2 | 80.3 ±0.2 | 69.9 ±0.5 | 74.7 ±0.9 | 64.1 ±1.4 | 59.2 ±0.0 | 29.9 ±0.2 |
| BSoftmax | 76.7 ±0.5 | 52.9 ±0.9 | 96.9 ±0.3 | 95.6 ±0.3 | 80.4 ±0.2 | 70.9 ±0.5 | 76.7 ±1.0 | 65.6 ±1.3 | 58.9 ±0.1 | 29.9 ±0.1 |
| SSP | 76.1 ±0.3 | 52.3 ±1.0 | 96.9 ±0.2 | 95.4 ±0.4 | 81.1 ±0.3 | 71.1 ±0.3 | 78.5 ±0.7 | 67.3 ±0.4 | 59.7 ±0.0 | 31.6 ±0.2 |
| CRT | 76.3 ±0.2 | 51.4 ±0.3 | 96.5 ±0.1 | 94.9 ±0.1 | 81.2 ±0.0 | 72.5 ±0.2 | 81.6 ±0.1 | 70.0 ±0.4 | 60.4 ±0.2 | 32.6 ±0.1 |
| BoDA$_r$ | 76.9 ±0.5 | 51.4 ±0.3 | 97.0 ±0.1 | 95.1 ±0.0 | 81.5 ±0.1 | 71.8 ±0.1 | 78.6 ±0.4 | 68.5 ±0.3 | 60.1 ±0.2 | 32.2 ±0.1 |
| BoDA-M$_r$ | 77.5 ±0.3 | 53.4 ±0.3 | 97.1 ±0.1 | 94.9 ±0.1 | 81.9 ±0.2 | 71.6 ±0.2 | 79.4 ±0.6 | 71.3 ±0.4 | 60.1 ±0.2 | 32.2 ±0.2 |
| BoDA$_{r,c}$ | 77.3 ±0.2 | 53.4 ±0.5 | 97.2 ±0.1 | 95.7 ±0.3 | 82.3 ±0.1 | 72.3 ±0.2 | 82.3 ±0.3 | 68.5 ±0.6 | 61.7 ±0.1 | 33.4 ±0.1 |
| BoDA-M$_{r,c}$ | 78.2 ±0.4 | 55.4 ±0.5 | 97.1 ±0.2 | 96.3 ±0.1 | 82.4 ±0.2 | 72.3 ±0.3 | 83.0 ±0.4 | 74.6 ±0.7 | 61.7 ±0.2 | 33.3 ±0.1 |
| Ours | **79.1** ±0.2 | **57.4** ±0.8 | **98.4** ±0.3 | **97.4** ±0.5 | **83.3** ±0.0 | **75.4** ±0.1 | **83.9** ±0.1 | 73.5 ±0.4 | **63.8** ±0.1 | **33.8** ±0.1 |

**Analysis.** Table 2 demonstrates the effectiveness of RC-Align in the challenging MDLT setting. Our method consistently outperforms specialized LT approaches. Specifically, RC-Align sets a new state-of-the-art on all benchmarks for average, and 4 out of 5 for worst-domain accuracy. The particularly strong performance in the **Worst** metric highlights the robustness of our framework; for instance, it achieves a notable gain of 2.9% on OfficeHome-MLT over the next best method and demonstrates superior robustness on the large-scale DomainNet-MLT as well. This empirical success confirms that RC-Align successfully learns representations that are robust not only to feature shifts ($P(X|Y)$) but also to severe and varying class imbalances (prior shifts). This validates our theoretical motivation: by directly optimizing a bound that accounts for both sources of divergence, RC-Align strikes a more effective balance than methods that primarily focus on either feature invariance or class re-balancing alone.

## 5.4 ANALYSIS AND VALIDATION

We conduct further analysis to validate the components of our framework and confirm our theoretical findings.

### 5.4.1 ABLATION STUDY

We analyze the contribution of the DA loss and the MAML framework, and Manifold Mixup on the VLCS dataset.

Table 3: Ablation study of RC-Align components on the **VLCS**. We start from an ERM baseline and incrementally add the meta-learning framework (MAML), the Domain-Class Distribution Alignment loss (DA Loss), and Manifold Mixup (M-Mixup). Accuracies are reported as percentages (%).

| Method | MAML | DA Loss | M-Mixup | Average Acc. | Worst Acc. |
|---|---|---|---|---|---|
| ERM (Baseline) | | | | 77.5 | 64.3 |
| MLDG (ERM + MAML) | ✓ | | | 77.2 | 65.2 |
| ERM + DA Loss | | ✓ | | 79.6 | 66.9 |
| RC-Align (w/o M-Mixup) | ✓ | ✓ | | 80.2 | 67.4 |
| **RC-Align (Full)** | ✓ | ✓ | ✓ | **81.0** | **68.5** |

**Impact of Components.** The results demonstrate the individual and synergistic contributions of each component. Both the DA Loss and the MAML framework individually improve upon the ERM baseline, showcasing their effectiveness in handling feature and domain shifts, respectively. Combining them in *RC-Align (w/o M-Mixup)* yields a significant performance boost, confirming

their complementary nature. The final addition of Manifold Mixup (M-Mixup) provides further regularization, leading to the best overall performance in both average and worst-domain accuracy. This validates the design of our full RC-Align model.

### 5.4.2 VERIFICATION OF THEORETICAL FINDINGS

Our theoretical analysis (Theorem 2) posits that the Domain-Class Distribution Alignment (DA) loss upper-bounds the feature mismatch term, implying that minimizing the DA loss should reduce the generalization gap. To empirically validate this, we performed experiments on the **VLCS** dataset using a Leave-One-Domain-Out (LODO) setup. For each scenario, we designated one domain as the target and the rest as source domains, then recorded the target's DA loss and the generalization gap (difference between target error and average source error) every 10 training steps.

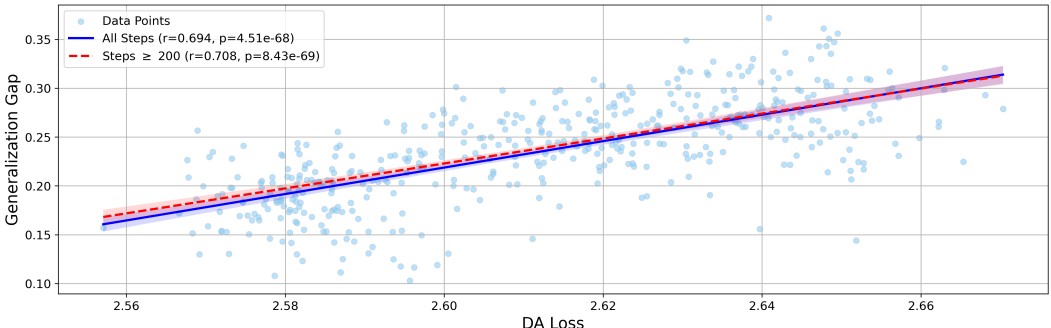

Figure 1: Empirical validation of Theorem 2 on **VLCS**. The scatter plot of DA Loss vs. Generalization Gap reveals a strong positive correlation across all steps (blue line; $r = 0.694, p < 10^{-67}$), which strengthens during the stabilized phase (steps $\geq 200$, red dashed line; $r = 0.708, p < 10^{-68}$).

**Observations:** Figure 1 empirically validates our theory, revealing a strong positive correlation between the DA loss and the generalization gap. This relationship notably strengthens during the stabilized training phase (steps $\geq 200$), suggesting that the DA loss becomes an even more precise indicator as features mature. This provides compelling evidence that the DA loss serves as a robust proxy for the feature mismatch term in our generalization bound (Eq. (3)).

## 6 CONCLUSION

In this work, we addressed the critical challenge of domain generalization under compound distribution shifts, where both marginal label distributions ($P(Y)$) and conditional feature distributions ($P(X|Y)$) diverge across domains. We introduced **RC-Align**, a novel meta-learning framework designed to explicitly counteract these dual challenges. The principled design of RC-Align is grounded in a new upper bound for domain generalization that decomposes the risk from unseen domains into distinct terms for **prior shift** and **feature shift**.

Our key theoretical contribution lies in proving that the practical Domain-Class Distribution Alignment (DA) loss, a core component of our method, serves as an **upper bound on the 1-Wasserstein distance**, which quantifies feature mismatch in our bound. This establishes a principled relationship between our training objective and the theoretical generalization gap. Furthermore, our analysis suggests that the meta-learning procedure helps minimize this bound by guiding source-domain alignment in a manner that improves performance on the target domain.

This theoretical foundation translates to strong empirical performance. RC-Align achieves state-of-the-art results not only on standard DG benchmarks but, more importantly, demonstrates superior robustness on challenging **Multi-Domain Long-Tailed Recognition (MDLT)** settings where compound shifts are most severe. Our analyses further validate this success, empirically confirming the correlation between the DA loss and the generalization gap predicted by our theory. By tightly integrating a practical meta-learning algorithm with a rigorous theoretical justification, RC-Align provides a principled approach for developing models that are robust to the complex, compound distribution shifts encountered in real-world deployments.

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

# A    DETAILED PROOFS FOR UPPER BOUND FOR DOMAIN GENERALIZATION

This appendix contains the detailed proofs for the theoretical results presented in Section 3, related to the upper bound for domain generalization and the DA loss analysis. We use the notation $\Theta = (\theta, \phi)$, d for the distance metric, and $\mathcal{D}_i$ for domains.

## A.1    PROOF OF THEOREM 1

**Theorem 1.** *[Query–support bound] Under* **??** *1, The risk on a query domain ($\mathcal{D}_i$) is bounded by the risk on the support mixture ($\mathcal{D}_{-i}$) plus mismatch terms:*

$$\mathcal{R}_{\mathcal{D}_i}(\Theta) \leq \mathcal{R}_{\mathcal{D}_{-i}}(\Theta) + \underbrace{\sum_{c \in \mathcal{C}} |\pi_i(c) - \pi_{-i}(c)| \cdot \mathcal{R}_{-i,c}(\Theta)}_{\text{Prior Shift}} + \underbrace{L_\ell \sum_{c \in \mathcal{C}} \pi_i(c) \, \mathsf{W}_1(P_{i,c}, P_{-i,c})}_{\text{Feature Shift}} , \quad (1)$$

*where $\mathsf{W}_1(\cdot, \cdot)$ $L_\ell$ are 1-Wasserstein distance and a lipschitz constant of $\ell$, respectively.*

*Proof.* Our goal is to bound the target risk $\mathcal{R}_{\mathcal{D}_i}(\Theta)$ using quantities related to the source mixture $\mathcal{R}_{\mathcal{D}_{-i}}(\Theta)$.

**Step 1: Decompose the Risk.** First, we express the total risk as the sum of per-class expected losses. We use the notation $\mathcal{R}_{\mathcal{D},c}(\Theta) = \mathbb{E}_{\mathbf{Z} \sim P_{\mathcal{D},c}}[\ell(g_\phi(\mathbf{Z}), c)]$.

$$\mathcal{R}_{\mathcal{D}_i}(\Theta) = \sum_{c \in \mathcal{C}} \pi_i(c) \, \mathcal{R}_{i,c}(\Theta)$$

$$\mathcal{R}_{\mathcal{D}_{-i}}(\Theta) = \sum_{c \in \mathcal{C}} \pi_{-i}(c) \, \mathcal{R}_{-i,c}(\Theta)$$

**Step 2: Isolate Distributional Shifts.** We add and subtract a hybrid term, $\sum_c \pi_i(c)\mathcal{R}_{-i,c}(\Theta)$, to separate the prior shift from the feature shift.

$$\mathcal{R}_{\mathcal{D}_i}(\Theta) - \mathcal{R}_{\mathcal{D}_{-i}}(\Theta)$$

$$= \sum_{c \in \mathcal{C}} (\pi_i(c)\mathcal{R}_{i,c}(\Theta) - \pi_{-i}(c)\mathcal{R}_{-i,c}(\Theta))$$

$$= \underbrace{\sum_{c \in \mathcal{C}} \pi_i(c) \, (\mathcal{R}_{i,c}(\Theta) - \mathcal{R}_{-i,c}(\Theta))}_{\text{Term A: Feature Shift}} + \underbrace{\sum_{c \in \mathcal{C}} (\pi_i(c) - \pi_{-i}(c)) \, \mathcal{R}_{-i,c}(\Theta)}_{\text{Term B: Prior Shift}}$$

**Step 3: Analyze the Prior Shift Term (Term B).** We bound Term B using the triangle inequality.

$$|\text{Term B}| \leq \sum_{c \in \mathcal{C}} |\pi_i(c) - \pi_{-i}(c)| \cdot \mathcal{R}_{-i,c}(\Theta)$$

**Step 4: Bound the Feature Shift Term (Term A).** To bound this term, we use the Kantorovich-Rubinstein Duality theorem. Since $\ell(g_\phi(z), c)$ is $L_\ell$-Lipschitz with respect to d, we can apply a scaled version of the theorem:

$$|\mathcal{R}_{i,c}(\Theta) - \mathcal{R}_{-i,c}(\Theta)| \leq L_\ell \cdot \mathsf{W}_1(P_{i,c}, P_{-i,c})$$

Now we can bound Term A:

$$|\text{Term A}| \leq \sum_{c \in \mathcal{C}} \pi_i(c) \cdot L_\ell \cdot \mathsf{W}_1(P_{i,c}, P_{-i,c})$$

**Step 5: Combine the Bounds.** Combining the bounds for Term A and Term B using the triangle inequality yields the final result. $\qquad\square$

## A.2 PROOF OF THEOREM 2

**Theorem 2.** *[DA loss controls the query–support gap] The total feature distribution mismatch (Feature Shift term) is upper-bounded by the DA loss:*

$$\sum_c \pi_i(c)\, \mathsf{W}_1\big(P_{i,c}, P_{-i,c}\big) \;\leq\; \mathbb{E}\,\mathcal{L}^{(i)}_{\mathrm{DA}}(\theta) \;+\; C^{(i)}_{\mathrm{scat}} \;+\; R, \tag{2}$$

*where $C^{(i)}_{\mathrm{scat}}$ is the average weighted scatter within the support domains (defined formally in Appendix C.2), and $R$ is the diameter of the feature space.*

*Proof.* This proof combines the results from the three lemmas detailed in Appendix C.2.

**Step 1: Average Over Classes.** We take the expectation of the inequality from Lemma 4 over all classes $c$ according to $\pi_i(c)$.

$$\sum_c \pi_i(c)\mathsf{W}_1(P_{i,c}, P_{-i,c}) \leq \underbrace{\sum_c \pi_i(c) \sum_{j\neq i} \omega_j\, \mathbb{E}_{\mathbf{Z}\sim P_{i,c}}\mathsf{d}(\mathbf{Z}, \mu_{j,c})}_{\text{Term 1}}$$

$$+ \underbrace{\sum_c \pi_i(c) \sum_{j\neq i} \omega_j S_{j,c}}_{\text{Term 2}}.$$

**Step 2: Relate Terms to Definitions.**

- **Term 1** is the definition of the positive transport cost $A_i(\theta)$ (Lemma 2).

- **Term 2** is the definition of $C^{(i)}_{\mathrm{scat}}$.

The inequality simplifies to:

$$\sum_c \pi_i(c)\mathsf{W}_1(P_{i,c}, P_{-i,c}) \leq A_i(\theta) + C^{(i)}_{\mathrm{scat}}.$$

**Step 3: Substitute the Bound on $A_i$.** We use the result from Lemma 3: $A_i(\theta) \leq \mathbb{E}\,\mathcal{L}^{(i)}_{\mathrm{DA}}(\theta) + R$. Substituting this gives the final result:

$$\sum_c \pi_i(c)\mathsf{W}_1(P_{i,c}, P_{-i,c}) \leq \mathbb{E}\,\mathcal{L}^{(i)}_{\mathrm{DA}}(\theta) + C^{(i)}_{\mathrm{scat}} + R.$$

$\square$

## A.3 PROOF OF THEOREM 3

**Theorem 3.** *[Mixture-aware target bound] For any model $\Theta$ and $\boldsymbol{\pi} \in \Delta^K$, the risk on $\mathcal{T}$ is bounded by:*

$$\mathcal{R}_{\mathcal{T}}(\Theta) \;\leq\; \sum_{i=1}^{K} \pi_i\, \mathcal{R}_{\mathcal{D}_i}(\Theta) \;+\; L_\ell\, \mathsf{W}_1\big(\mathcal{T}, \widetilde{\mathcal{T}}_{\boldsymbol{\pi}}\big)\,. \tag{4}$$

*Proof.* The proof relies on the Lipschitz continuity of the risk functional and the linearity of expectation over mixtures.

**Step 1: Risk as a Lipschitz Functional.** The expected risk $\mathcal{R}_{\mathcal{D}}(\Theta)$ is a functional of the distribution $\mathcal{D}$. Because the loss $\ell$ is $L_\ell$-Lipschitz w.r.t. $\mathsf{d}$, the risk functional is $L_\ell$-Lipschitz w.r.t. $\mathsf{W}_1$. By the Kantorovich-Rubinstein duality:

$$|\mathcal{R}_{\mathcal{T}}(\Theta) - \mathcal{R}_{\widetilde{\mathcal{T}}_{\boldsymbol{\pi}}}(\Theta)| \leq L_\ell\, \mathsf{W}_1(\mathcal{T}, \widetilde{\mathcal{T}}_{\boldsymbol{\pi}}).$$

Rearranging to isolate the target risk:

$$\mathcal{R}_{\mathcal{T}}(\Theta) \le \mathcal{R}_{\widetilde{\mathcal{T}}_{\boldsymbol{\pi}}}(\Theta) + L_\ell \, \mathsf{W}_1(\mathcal{T}, \widetilde{\mathcal{T}}_{\boldsymbol{\pi}}).$$

**Step 2: Linearity of Risk over Mixtures.** By definition, the expectation over a mixture distribution is the weighted average of the expectations over its components.

$$\mathcal{R}_{\widetilde{\mathcal{T}}_{\boldsymbol{\pi}}}(\Theta) = \mathbb{E}_{(\mathbf{X},\mathbf{Y}) \sim \sum_i \pi_i \mathcal{D}_i}[\ell(\dots)] = \sum_{i=1}^{K} \pi_i \mathcal{R}_{\mathcal{D}_i}(\Theta).$$

**Step 3: Combining the Results.** Substituting the result from Step 2 back into Step 1 completes the proof. $\qquad\square$

# B    Detailed Proofs for MAML Analysis

This appendix contains the detailed proofs for the theoretical results presented in Section 3.3, focusing on the optimization dynamics.

## B.1    Proof of Lemma 1

**Lemma 1.** *[One-Step Contraction] Under standard smoothness/PL conditions, the inner adaptation step yields a geometric contraction of the objective:*

$$J_{-i}(\Theta_i^+) - J_{-i}^\star \le \rho(\alpha)\big(J_{-i}(\Theta) - J_{-i}^\star\big), \qquad (5)$$

*where the contraction factor $\rho(\alpha) < 1$.*

*Proof.* We aim to show that the inner objective function value contracts towards the optimum $J_{-i}^\star$ under smoothness and the PL condition (Assumption 2).

**Step 1: Apply the Descent Lemma (L-smoothness).** Since $J_{-i}(\Theta)$ is $L$-smooth, it satisfies the Descent Lemma:

$$J_{-i}(\Theta_i^+) \le J_{-i}(\Theta) + \langle \nabla J_{-i}(\Theta), \Theta_i^+ - \Theta \rangle + \frac{L}{2}\|\Theta_i^+ - \Theta\|^2.$$

**Step 2: Substitute the Gradient Descent Update Rule.** The inner adaptation step is $\Theta_i^+ = \Theta - \alpha \nabla J_{-i}(\Theta)$. Substituting this into the inequality from Step 1:

$$J_{-i}(\Theta_i^+) \le J_{-i}(\Theta) - \alpha\|\nabla J_{-i}(\Theta)\|^2 + \frac{L\alpha^2}{2}\|\nabla J_{-i}(\Theta)\|^2$$

$$= J_{-i}(\Theta) - \alpha\left(1 - \frac{L\alpha}{2}\right)\|\nabla J_{-i}(\Theta)\|^2.$$

**Step 3: Apply the Polyak–Lojasiewicz (PL) Condition.** The learning rate condition $0 < \alpha < 2/L$ ensures the coefficient $\alpha(1 - \frac{L\alpha}{2})$ is positive. We utilize the PL condition:

$$\|\nabla J_{-i}(\Theta)\|^2 \ge 2\mu\big(J_{-i}(\Theta) - J_{-i}^\star\big).$$

Substituting this lower bound:

$$J_{-i}(\Theta_i^+) \le J_{-i}(\Theta) - \alpha\left(1 - \frac{L\alpha}{2}\right) \cdot 2\mu\big(J_{-i}(\Theta) - J_{-i}^\star\big).$$

**Step 4: Rearrange to Show Contraction.** We rearrange the terms to isolate the optimization gap at $\Theta_i^+$:

$$J_{-i}(\Theta_i^+) - J_{-i}^\star \le J_{-i}(\Theta) - J_{-i}^\star - 2\mu\alpha\left(1 - \frac{L\alpha}{2}\right)\big(J_{-i}(\Theta) - J_{-i}^\star\big)$$

$$= \left(1 - 2\mu\alpha\left(1 - \frac{L\alpha}{2}\right)\right)\big(J_{-i}(\Theta) - J_{-i}^\star\big).$$

Defining the contraction factor $\rho(\alpha) := 1 - 2\mu\alpha(1 - \frac{L\alpha}{2})$, we obtain the result. Since the subtracted term is positive, $\rho(\alpha) < 1$. $\qquad\square$

## B.2 Proof of Proposition 1

**Proposition 1.** *[Reduction of the DA alignment term] The DA loss on the source domains after adaptation ($\Theta_i^+$) is bounded as:*

$$\mathbb{E}\mathcal{L}_{\mathrm{DA}}^{(-i)}(\Theta_i^+) \leq \frac{\rho(\alpha)}{\lambda_{\mathrm{DA}}} J_{-i}(\Theta) + \frac{1 - \rho(\alpha)}{\lambda_{\mathrm{DA}}} J_{-i}^\star. \tag{6}$$

*Proof.* We relate the contraction of the total objective $J_{-i}$ to the reduction of the DA loss component $\mathbb{E}\mathcal{L}_{\mathrm{DA}}^{(-i)}$.

**Step 1: Bound DA Loss by the Total Objective.** Recall the definition of the inner objective $J_{-i}(\Theta)$:

$$J_{-i}(\Theta) = \mathcal{R}_{\mathcal{D}_{-i}}(\Theta) + \lambda_{\mathrm{DA}}\mathbb{E}\mathcal{L}_{\mathrm{DA}}^{(-i)}(\theta).$$

Assuming the classification risk $\mathcal{R}_{\mathcal{D}_{-i}}(\Theta)$ is non-negative, the DA loss term is upper-bounded by the total objective:

$$\lambda_{\mathrm{DA}}\mathbb{E}\mathcal{L}_{\mathrm{DA}}^{(-i)}(\theta) \leq J_{-i}(\Theta).$$

This holds for the post-adaptation parameters $\Theta_i^+ = (\theta_i^+, \phi_i^+)$ as well:

$$\mathbb{E}\mathcal{L}_{\mathrm{DA}}^{(-i)}(\theta_i^+) \leq \frac{1}{\lambda_{\mathrm{DA}}} J_{-i}(\Theta_i^+).$$

**Step 2: Apply the Contraction Result.** From Lemma 1:

$$J_{-i}(\Theta_i^+) - J_{-i}^\star \leq \rho(\alpha)\big(J_{-i}(\Theta) - J_{-i}^\star\big).$$

Rearranging this inequality to bound $J_{-i}(\Theta_i^+)$:

$$J_{-i}(\Theta_i^+) \leq \rho(\alpha)J_{-i}(\Theta) - \rho(\alpha)J_{-i}^\star + J_{-i}^\star$$
$$= \rho(\alpha)J_{-i}(\Theta) + (1 - \rho(\alpha))J_{-i}^\star.$$

**Step 3: Combine the Bounds.** Substituting the upper bound for $J_{-i}(\Theta_i^+)$ from Step 2 into the inequality from Step 1:

$$\mathbb{E}\mathcal{L}_{\mathrm{DA}}^{(-i)}(\theta_i^+) \leq \frac{1}{\lambda_{\mathrm{DA}}} \left(\rho(\alpha)J_{-i}(\Theta) + (1 - \rho(\alpha))J_{-i}^\star\right).$$

$$\mathbb{E}\mathcal{L}_{\mathrm{DA}}^{(-i)}(\theta_i^+) \leq \frac{\rho(\alpha)}{\lambda_{\mathrm{DA}}} J_{-i}(\Theta) + \frac{1 - \rho(\alpha)}{\lambda_{\mathrm{DA}}} J_{-i}^\star.$$

$\square$

## C   Extended Details on Theory and Methodology

This appendix provides the detailed definitions, assumptions, lemmas, and discussions that were condensed in the main paper (Sections 3 and 4).

### C.1   Detailed Setup and Assumptions (Section 3)

**Lipschitz Continuity and Boundedness.** We assume the loss function $\ell$ is bounded and Lipschitz continuous with respect to the distance metric d in the feature space $\mathcal{Z}$:

$$0 \leq \ell(g_\phi(z), c) \leq B_\ell, \qquad \big|\ell(g_\phi(z), c) - \ell(g_\phi(z'), c)\big| \leq L_\ell\, \mathsf{d}(z, z') \quad \forall z, z' \in \mathcal{Z}. \tag{11}$$

**Feature Space Diameter.** We assume the feature space $\mathcal{Z}$ is compact, implying a finite diameter bound:

$$R = \sup_{z, z' \in \mathcal{Z}} \mathsf{d}(z, z') < \infty. \tag{12}$$

**Class-Conditional Risk.** We define the class-conditional expected risk for a domain $\mathcal{D}$ and class $c$ as $\mathcal{R}_{\mathcal{D},c}(\Theta) = \mathbb{E}_{\mathbf{Z} \sim P_{\mathcal{D},c}}[\ell(g_\phi(\mathbf{Z}), c)]$.

## C.2  Detailed Lemmas for Theorem 2 (Section 3.1)

The proof of Theorem 2 relies on the following three lemmas, which connect the Wasserstein distance to the DA loss via an InfoNCE decomposition and a two-step transport argument. All proofs in these lemmas are provided in Lemmas 2 to 4.

**Lemma 2.** *[InfoNCE-style identity] The expected Domain-Class Distribution Alignment loss for the query domain $\mathcal{D}_i$, denoted $\mathbb{E}\,\mathcal{L}_{\mathrm{DA}}^{(i)}(\theta)$, can be decomposed as:*

$$\mathbb{E}\,\mathcal{L}_{\mathrm{DA}}^{(i)}(\theta) \;=\; A_i(\theta) \;+\; \mathbb{E}\Phi(\mathbf{Z};\theta), \tag{13}$$

*where $A_i(\theta)$ is the expected distance from a query feature to its corresponding positive centroids $\{\mu_{j,c}(\theta)\}$ in the support mixture (the "positive transport cost"):*

$$A_i(\theta) := \sum_c \pi_i(c) \sum_{j \neq i} \omega_j \, \mathbb{E}_{\mathbf{Z} \sim P_{i,c}} \mathsf{d}(f_\theta(\mathbf{Z}), \mu_{j,c}(\theta)).$$

*and $\Phi(\mathbf{Z};\theta)$ is a term related to the negative centroids (the log-partition function of the negatives).*

Lemma 2 shows that minimizing the DA loss simultaneously minimizes the positive transport cost $A_i$ and the negative term $\Phi(\mathbf{Z})$. To isolate $A_i$, we need to bound $\Phi(\mathbf{Z})$.

**Lemma 3.** *[Lower bound on $\Phi$ by diameter] The term $\Phi(\mathbf{Z};\theta)$ is lower-bounded using the diameter $R$ (Eq. 12). This leads to a crucial consequence: the expected positive transport cost $A_i$ is upper-bounded by the measurable DA loss.*

$$A_i(\theta) \leq \mathbb{E}\,\mathcal{L}_{\mathrm{DA}}^{(i)}(\theta) + R.$$

Now that we have connected the DA loss to the positive transport cost $A_i$ (which measures the distance from query features to support centroids), we must relate $A_i$ back to the Wasserstein distance between the underlying distributions ($P_{i,c}$ and $P_{-i,c}$).

Let $\bar{\mu}_{-i,c} := \sum_{j \neq i} \omega_j \, \delta_{\mu_{j,c}}$ be the mixture of support-domain centroids for class $c$. Let $S_{j,c} := \mathbb{E}_{\mathbf{Z} \sim P_{j,c}} \mathsf{d}(f_\theta(\mathbf{Z}), \mu_{j,c})$ be the average intra-domain spread (scatter) of features.

**Lemma 4.** *[Two-step transport] For any class $c$, the Wasserstein distance between $P_{i,c}$ and $P_{-i,c}$ can be bounded by applying the triangle inequality, using the centroids as intermediates:*

$$\mathsf{W}_1\big(P_{i,c}, P_{-i,c}\big) \;\leq\; \underbrace{\mathsf{W}_1\big(P_{i,c}, \bar{\mu}_{-i,c}\big)}_{\leq \sum_{j \neq i} \omega_j \, \mathbb{E}_{\mathbf{Z} \sim P_{i,c}} \mathsf{d}(f_\theta(\mathbf{Z}), \mu_{j,c})} \;+\; \underbrace{\mathsf{W}_1\big(\bar{\mu}_{-i,c}, P_{-i,c}\big)}_{\leq \sum_{j \neq i} \omega_j \, S_{j,c}}.$$

**Definition of $C_{\mathrm{scat}}$.** Theorem 2 combines these results. The term $C_{\mathrm{scat}}^{(i)}$ mentioned in the theorem is derived from the second term of Lemma 4, averaged over the classes according to the query prior $\pi_i(c)$:

$$C_{\mathrm{scat}}^{(i)} := \sum_{c \in \mathcal{C}} \pi_i(c) \sum_{j \neq i} \omega_j S_{j,c}.$$

## C.3  Proof of Lemma 2

**Lemma 2.** *[InfoNCE-style identity] The expected Domain-Class Distribution Alignment loss for the query domain $\mathcal{D}_i$, denoted $\mathbb{E}\,\mathcal{L}_{\mathrm{DA}}^{(i)}(\theta)$, can be decomposed as:*

$$\mathbb{E}\,\mathcal{L}_{\mathrm{DA}}^{(i)}(\theta) \;=\; A_i(\theta) \;+\; \mathbb{E}\Phi(\mathbf{Z};\theta), \tag{13}$$

*where $A_i(\theta)$ is the expected distance from a query feature to its corresponding positive centroids $\{\mu_{j,c}(\theta)\}$ in the support mixture (the "positive transport cost"):*

$$A_i(\theta) := \sum_c \pi_i(c) \sum_{j \neq i} \omega_j \, \mathbb{E}_{\mathbf{Z} \sim P_{i,c}} \mathsf{d}(f_\theta(\mathbf{Z}), \mu_{j,c}(\theta)).$$

*and $\Phi(\mathbf{Z};\theta)$ is a term related to the negative centroids (the log-partition function of the negatives).*

*Proof.* The proof relies on analyzing the structure of the InfoNCE loss for a single sample and then taking the expectation. We use $s_{j,c}(z) = \mathsf{d}(z, \mu_{j,c})$ for brevity, noting the dependency on $\theta$.

**Step 1: Single-Sample Identity.** Let's analyze the DA loss for a single sample $(z, c)$ from domain $i$. We define the negative partition function $\Phi(z)$:

$$\Phi(z) = \log \sum_{(j',c') \neq (i,c)} e^{-s_{j',c'}(z)}.$$

Using the logarithm property $\log(a/b) = \log(a) - \log(b)$, we expand the loss (assuming $K - 1$ support domains and uniform weights $\omega_j = \frac{1}{K-1}$):

$$\text{Loss}(z, c, i) = \sum_{j \neq i} \omega_j \left( - \log \left( e^{-s_{j,c}(z)} \right) + \log \left( \sum_{(j',c') \neq (i,c)} e^{-s_{j',c'}(z)} \right) \right)$$

$$= \sum_{j \neq i} \omega_j \left( s_{j,c}(z) + \Phi(z) \right)$$

Since $\Phi(z)$ does not depend on $j$ and $\sum_{j \neq i} \omega_j = 1$:

$$\text{Loss}(z, c, i) = \sum_{j \neq i} \omega_j s_{j,c}(z) + \Phi(z).$$

**Step 2: Taking the Expectation.** Now, we take the expectation over samples $(\mathbf{Z}, \mathbf{C})$ drawn from the query domain distribution $\mathcal{D}_i$.

$$\mathbb{E} \, \mathcal{L}_{\text{DA}}^{(i)}(\theta) = \mathbb{E} \left[ \sum_{j \neq i} \omega_j s_{j,\mathbf{C}}(\mathbf{Z}) \right] + \mathbb{E}[\Phi(\mathbf{Z})].$$

Let's analyze the first term on the RHS. By definition of expectation over $\mathcal{D}_i$:

$$\mathbb{E} \left[ \sum_{j \neq i} \omega_j s_{j,\mathbf{C}}(\mathbf{Z}) \right] = \sum_{c \in \mathcal{C}} \pi_i(c) \mathbb{E}_{\mathbf{Z} \sim P_{i,c}} \left[ \sum_{j \neq i} \omega_j \mathsf{d}(\mathbf{Z}, \mu_{j,c}) \right]$$

$$= \sum_{c \in \mathcal{C}} \pi_i(c) \sum_{j \neq i} \omega_j \mathbb{E}_{\mathbf{Z} \sim P_{i,c}}[\mathsf{d}(\mathbf{Z}, \mu_{j,c})]$$

This expression matches the definition of $A_i(\theta)$. Substituting this back, we get the desired identity. $\square$

## C.4 PROOF OF LEMMA 3

**Lemma 3.** *[Lower bound on $\Phi$ by diameter] The term $\Phi(\mathbf{Z}; \theta)$ is lower-bounded using the diameter $R$ (Eq. 12). This leads to a crucial consequence: the expected positive transport cost $A_i$ is upper-bounded by the measurable DA loss.*

$$A_i(\theta) \leq \mathbb{E} \, \mathcal{L}_{\text{DA}}^{(i)}(\theta) + R.$$

*Proof.* The proof involves bounding $\Phi(z)$ using the diameter $R$ and then relating $A_i$ back to $\mathcal{L}_{\text{DA}}$.

**Step 1: Bounding $\Phi(z)$.**

1. **Bound the Distance:** Since $z$ and $\mu_{j',c'}$ are in $\mathcal{Z}$, their distance is bounded by $R$.
$$s_{j',c'}(z) = \mathsf{d}(z, \mu_{j',c'}) \leq R.$$

2. **Apply Exponential and Sum:** $e^{-s_{j',c'}(z)} \geq e^{-R}$. Let $N$ be the number of terms in the sum (number of negative pairs).
$$\sum_{(j',c') \neq (i,c)} e^{-s_{j',c'}(z)} \geq N \cdot e^{-R}.$$

3. **Apply Logarithm:**

$$\Phi(z) = \log\left(\sum_{(j',c')\neq(i,c)} e^{-s_{j',c'}(z)}\right) \geq \log(N \cdot e^{-R}) = \log N - R.$$

**Step 2: Bounding $A_i$.**

1. **Recall Identity (Lemma 2):**

$$A_i(\theta) = \mathbb{E}\,\mathcal{L}_{\mathrm{DA}}^{(i)}(\theta) - \mathbb{E}[\Phi(\mathbf{Z};\theta)].$$

2. **Apply the Bound on $\Phi(\mathbf{Z})$:** We have $\Phi(\mathbf{Z}) \geq \log N - R$, so $-\Phi(\mathbf{Z}) \leq R - \log N$.

3. **Substitute and Conclude:**

$$A_i(\theta) \leq \mathbb{E}\,\mathcal{L}_{\mathrm{DA}}^{(i)}(\theta) + R - \log N.$$

Since $N \geq 1$, $\log N \geq 0$. Removing this non-positive term yields the simpler bound:

$$A_i(\theta) \leq \mathbb{E}\,\mathcal{L}_{\mathrm{DA}}^{(i)}(\theta) + R.$$

$\square$

### C.5 PROOF OF LEMMA 4

**Lemma 4.** *[Two-step transport] For any class $c$, the Wasserstein distance between $P_{i,c}$ and $P_{-i,c}$ can be bounded by applying the triangle inequality, using the centroids as intermediates:*

$$\mathsf{W}_1(P_{i,c}, P_{-i,c}) \leq \underbrace{\mathsf{W}_1(P_{i,c}, \bar{\mu}_{-i,c})}_{\leq \sum_{j\neq i}\omega_j\,\mathbb{E}_{\mathbf{Z}\sim P_{i,c}}\mathsf{d}(f_\theta(\mathbf{Z}),\mu_{j,c})} + \underbrace{\mathsf{W}_1(\bar{\mu}_{-i,c}, P_{-i,c})}_{\leq \sum_{j\neq i}\omega_j\,S_{j,c}}.$$

*Proof.* The proof proceeds by applying the triangle inequality and bounding the resulting terms.

**Step 1: Triangle Inequality.** The 1-Wasserstein distance satisfies the triangle inequality. We introduce the mixture of centroids, $\bar{\mu}_{-i,c}$, as an intermediate point.

$$\mathsf{W}_1(P_{i,c}, P_{-i,c}) \leq \mathsf{W}_1(P_{i,c}, \bar{\mu}_{-i,c}) + \mathsf{W}_1(\bar{\mu}_{-i,c}, P_{-i,c}).$$

**Step 2: Bounding the First Term ($P_{i,c} \to \bar{\mu}_{-i,c}$).** The Wasserstein distance is the minimum cost over all transport plans. We construct a specific plan: transport $\mathbf{Z} \sim P_{i,c}$ to $\mu_{j,c}$ with probability $\omega_j$. The expected cost is:

$$\mathrm{Cost} = \mathbb{E}_{\mathbf{Z}\sim P_{i,c}}\left[\sum_{j\neq i}\omega_j\mathsf{d}(\mathbf{Z}, \mu_{j,c})\right] = \sum_{j\neq i}\omega_j\mathbb{E}_{\mathbf{Z}\sim P_{i,c}}[\mathsf{d}(\mathbf{Z}, \mu_{j,c})].$$

$\mathsf{W}_1(P_{i,c}, \bar{\mu}_{-i,c})$ is upper bounded by this cost.

**Step 3: Bounding the Second Term ($\bar{\mu}_{-i,c} \to P_{-i,c}$).** Using the convexity of the Wasserstein distance:

$$\mathsf{W}_1(\bar{\mu}_{-i,c}, P_{-i,c}) \leq \sum_{j\neq i}\omega_j\mathsf{W}_1(\delta_{\mu_{j,c}}, P_{j,c}).$$

The term $\mathsf{W}_1(\delta_{\mu_{j,c}}, P_{j,c})$ is the transport cost from a point mass at $\mu_{j,c}$ to the distribution $P_{j,c}$, which is exactly the intra-domain spread $S_{j,c}$.

$$\mathsf{W}_1(\delta_{\mu_{j,c}}, P_{j,c}) = \mathbb{E}_{\mathbf{Z}\sim P_{j,c}}[\mathsf{d}(\mathbf{Z}, \mu_{j,c})] = S_{j,c}.$$

Thus, $\mathsf{W}_1(\bar{\mu}_{-i,c}, P_{-i,c}) \leq \sum_{j\neq i}\omega_j S_{j,c}$.

**Step 4: Combining the Bounds.** Substituting the bounds from Step 2 and Step 3 back into Step 1 completes the proof. $\square$

## C.6 THE FINAL COMBINED BOUND (SECTION 3.2)

We combine the Mixture-aware bound (Theorem 3) and the final theoretical bound (Eq. (3)) to yield a single, comprehensive guarantee for an arbitrary target domain $\mathcal{T}$.

$$\mathcal{R}_{\mathcal{T}}(\Theta) \leq \sum_i \pi_i \mathcal{R}_{\mathcal{D}_i}(\Theta) + L_\ell \, \mathsf{W}_1\big(\mathcal{T}, \widetilde{\mathcal{T}}_{\boldsymbol{\pi}}\big)$$

$$\leq \sum_i \pi_i \left\{ \underbrace{\mathcal{R}_{\mathcal{D}_{-i}}(\Theta)}_{\text{Support Error}} + \underbrace{\sum_c |\pi_i(c) - \pi_{-i}(c)| \mathcal{R}_{-i,c}(\Theta)}_{\text{Prior Shift Impact}} + \underbrace{L_\ell\big(\mathbb{E}\,\mathcal{L}_{\text{DA}}^{(i)}(\theta) + C_{\text{scat}}^{(i)} + R\big)}_{\text{Feature Shift}} \right\}$$

$$+ L_\ell \, \mathsf{W}_1\big(\mathcal{T}, \widetilde{\mathcal{T}}_{\boldsymbol{\pi}}\big).$$

This final inequality connects the performance on an arbitrary, unseen target domain $\mathcal{T}$ to quantities measurable on the source domains, explicitly capturing how the algorithm controls both prior shift (by minimizing $\mathcal{R}_{-i,c}(\Theta)$) and feature shift (by minimizing the DA loss).

## C.7 DETAILED MAML ANALYSIS AND DISCUSSION (SECTION 3.3)

**Setup for Analysis.** The population inner objective $J_{-i}(\Theta)$ is defined as:

$$J_{-i}(\Theta) := \mathcal{R}\mathcal{D}_{-i}(\Theta) + \lambda_{\text{DA}}\mathbb{E}\mathcal{L}_{\text{DA}}^{(-i)}(\theta). \tag{14}$$

**Assumption 2** (Smoothness and Polyak–Lojasiewicz (PL) Condition)**.** *$J_{-i}(\Theta)$ is $L$-smooth and satisfies the PL inequality with parameter $\mu > 0$. Furthermore, the learning rate satisfies $0 < \alpha < 2/L$.*

*Remark* 1 (Discussion on the PL Condition)*.* The PL condition is a strong assumption, particularly for highly non-convex landscapes typical of deep neural networks. While this assumption may not hold globally in practice, it is a standard tool in optimization analysis used to establish convergence rates (Karimi et al., 2016). Furthermore, recent studies suggest that overparameterized networks may satisfy conditions similar to PL in relevant regions of the parameter space (Liu et al., 2022). We adopt this assumption to provide theoretical insights into the local optimization dynamics of the inner loop.

*Remark* 2 (Manifold Mixup and Optimization Dynamics)*.* The integration of Manifold Mixup in the inner loop serves as a vital regularization technique that potentially smooths the optimization landscape. By encouraging linear behavior in the feature space between source domains, Mixup may facilitate the contraction analyzed in Lemma 1, potentially improving the effective smoothness constant $L$ and leading to more stable adaptation.

*Remark* 3 (The Connection between Source Alignment and Target Generalization)*.* The structure of the MLDG-style meta-learning procedure is key to its effectiveness. The inner loop performs adaptation on the source domains ($-i$), actively reducing the source alignment objective $J_{-i}$ (Proposition 1). The outer loop then minimizes the meta-objective $\min_\Theta \mathbb{E}_i[\mathcal{R}\mathcal{D}_i(\Theta_i^+)]$, evaluating the adapted parameters on the held-out target domain ($i$). This optimization forces the model to find an initialization $\Theta$ such that the gradient step towards better source alignment simultaneously leads to better target generalization.

## C.8 POPULATION LOSS DEFINITIONS (SECTION 4.1)

We provide the population definitions for the loss functions used in the theoretical analysis.

**Classification Loss (Population).**

$$\mathcal{L}_{\text{CE}}(\theta, \phi) = \mathbb{E}_{(\mathbf{X},\mathbf{Y})}\big[\text{CE}(g_\phi(f_\theta(\mathbf{X})), \mathbf{Y})\big]. \tag{15}$$

**Domain-Class Distribution Alignment (DA) Loss (Population).** Let $\mu_{j,c}(\theta)$ be the centroid of class $c$ in domain $j$. The objective for domain $i$ is:

$$\mathcal{L}_{\text{DA}}^{(i)}(\theta) = \mathbb{E}_{(\mathbf{X},c)\sim\mathcal{D}_i}\left[\frac{-1}{K-1}\sum_{j\neq i}\log\frac{\exp\big(-\mathsf{d}(f_\theta(\mathbf{X}), \mu_{j,c}(\theta))\big)}{\sum_{(j',c')\neq(i,c)}\exp\big(-\mathsf{d}(f_\theta(\mathbf{X}), \mu_{j',c'}(\theta))\big)}\right]. \tag{16}$$

## C.9 ERM INSUFFICIENCY AND NECESSITY OF BOUNDS

A crucial implication of this analysis is the insufficiency of standard Empirical Risk Minimization (ERM), even when the target domain lies within the convex hull of the source domains. Theorem 3 bounds the target risk $R_T(\Theta)$ by the Source Risk ($\sum \pi_i R_{D_i}(\Theta)$) and the Distribution Shift term ($L_l W_1(T, T_\pi)$). ERM only minimizes the Source Risk.

Critically, these two terms are coupled via the learned representation $\Theta$. The Shift Term is measured in the feature space and is therefore optimizable. However, ERM often minimizes Source Risk by learning source-specific shortcuts (e.g., spurious correlations Izmailov et al. (2022); Chen et al. (2023); Deng et al. (2023)), which paradoxically increases the Shift Term. Therefore, controlling only the risk does not guarantee low target risk.

Furthermore, the Shift Term in Theorem 3 is intractable because it depends on the unknown target distribution $T$. Theorems 1 and 2 are essential as they derive the DA loss as a tractable surrogate for this intractable shift term. Our meta-learning framework operationalizes this bound by treating each source domain ($\mathcal{D}_i$) as a virtual target. Unlike ERM, which optimizes for the source mixture, our approach optimizes for generalization across simulated unseen tasks.

# D ALGORITHM DETAILS

This appendix provides the detailed pseudocode for the RC-Align training procedure (Algorithm 1) and the computation of the DA loss (Algorithm 2).

---

**Algorithm 1** RC-Align (Robust Compound Alignment) Training Procedure

---

1: **Input:** Source domains $\mathbb{D} = \{\mathcal{D}_1, \ldots, \mathcal{D}_K\}$, learning rates $\alpha$ (inner), $\eta$ (outer), $\lambda_{\text{DA}}$, Mixup parameter $\alpha_{\text{mixup}}$.
2: **Initialize:** Model parameters $\Theta = (\theta, \phi)$.
3: **while** not converged **do**
4:      Sample a mini-batch $B_k$ from each domain $\mathcal{D}_k$.
5:      **for** $i = 1$ to $K$ **do**                 $\triangleright$ LODO Episode: $i$ is the target domain
6:          Compute support centroids $\{\mu\}_{-i}$ using $B_{-i} = \{B_k\}_{k \neq i}$ (w/o grad).
7:          Initialize inner loss accumulator: $\mathcal{L}_{\text{inner}} = 0$.
8:          Initialize adapted parameters: $\Theta' = \Theta$.
9:          # — Inner (Adaptation) Step with Manifold Mixup —
10:         $N_{\text{pairs}} = 0$.
11:         **for** $(j_1, j_2)$ in RandomPairs($\{k\}_{k \neq i}$) **do**
12:             $N_{\text{pairs}} \leftarrow N_{\text{pairs}} + 1$.
13:             Get batches $(X_{j_1}, Y_{j_1}), (X_{j_2}, Y_{j_2})$.
14:             $Z_{j_1} \leftarrow f_\theta(X_{j_1})$
15:             $Z_{j_2} \leftarrow f_\theta(X_{j_2})$
16:             Sample mixing coefficient $\lambda \sim \text{Beta}(\alpha_{\text{mixup}}, \alpha_{\text{mixup}})$.
17:             $Z_{\text{mix}} \leftarrow \lambda Z_{j_1} + (1 - \lambda) Z_{j_2}$.
18:             # Calculate mixed composite loss
19:             $\mathcal{L}_{\text{CE},j_1} \leftarrow \widehat{\mathcal{L}}_{\text{CE}}(Z_{\text{mix}}, Y_{j_1}; \phi)$
20:             $\mathcal{L}_{\text{DA},j_1} \leftarrow \widehat{\mathcal{L}}_{\text{DA}}(Z_{\text{mix}}, Y_{j_1}, d_{j_1}; \theta, \{\mu\}_{-i})$
21:             $\widehat{\mathcal{L}}_{j_1} \leftarrow \mathcal{L}_{\text{CE},j_1} + \lambda_{\text{DA}} \mathcal{L}_{\text{DA},j_1}$
22:             $\mathcal{L}_{\text{CE},j_2} \leftarrow \widehat{\mathcal{L}}_{\text{CE}}(Z_{\text{mix}}, Y_{j_2}; \phi)$
23:             $\mathcal{L}_{\text{DA},j_2} \leftarrow \widehat{\mathcal{L}}_{\text{DA}}(Z_{\text{mix}}, Y_{j_2}, d_{j_2}; \theta, \{\mu\}_{-i})$
24:             $\widehat{\mathcal{L}}_{j_2} \leftarrow \mathcal{L}_{\text{CE},j_2} + \lambda_{\text{DA}} \mathcal{L}_{\text{DA},j_2}$
25:             $\mathcal{L}_{\text{mix}} \leftarrow \lambda \widehat{\mathcal{L}}_{j_1} + (1 - \lambda) \widehat{\mathcal{L}}_{j_2}$
26:             $\mathcal{L}_{\text{inner}} \leftarrow \mathcal{L}_{\text{inner}} + \mathcal{L}_{\text{mix}}$.
27:         **end for**
28:         $\mathcal{L}_{\text{inner}} \leftarrow \mathcal{L}_{\text{inner}} / N_{\text{pairs}}$.
29:         $\Theta' \leftarrow \Theta - \alpha \nabla_\Theta \mathcal{L}_{\text{inner}}$                 $\triangleright$ Adaptation (FOMAML)
30:         # — Outer (Meta) Step —
31:         $\mathcal{L}_{\text{outer}} \leftarrow \widehat{\mathcal{L}}_{\text{CE}}(B_i; \Theta')$             $\triangleright$ Evaluate on target domain
32:         Update $\Theta$ using $\nabla_\Theta \mathcal{L}_{\text{outer}}$ with optimizer step $\eta$.
33:      **end for**
34: **end while**

---

---

**Algorithm 2** Function: ComputeDALoss (Aligned with Implementation)

---

**Require:** Features $Z$, Labels $Y$, Domain $D_Z$, Centroids $\mathcal{C}$ (pre-normalized), Temperature $T$.

1: // Normalize input features (as implemented in the Python code)
2: $Z^{norm} \leftarrow \text{Normalize}(Z)$
3: // Use negative L2 distance as the similarity score (logit)
4: $Logits \leftarrow -\text{PairwiseL2Distance}(\mathcal{C}, Z^{norm})$
5: $Logits \leftarrow Logits/T$
6: // Contrastive Loss (InfoNCE style)
7: // Define Positive Mask $M_{pos}$
8: $M_{class} \leftarrow (\text{Labels}(\mathcal{C}) == Y)$
9: $M_{domain} \leftarrow (\text{Domains}(\mathcal{C}) == D_Z)$
10: $M_{pos} \leftarrow M_{class} \text{ AND } (\neg M_{domain})$ ▷ Same class AND different domain
11: // Define Negative Logits (Mask out positives for the denominator)
12: $L_{neg} \leftarrow Logits.$ Set $L_{neg}[M_{pos}] = -\infty.$
13: // Calculate LogSumExp over negative examples
14: $LSE \leftarrow \text{LogSumExp}(L_{neg})$
15: $LogProbs \leftarrow Logits - LSE$
16: // Calculate mean negative log probability of positive examples
17: $\mathcal{L}_{DA} \leftarrow -\text{Mean}(LogProbs[M_{pos}])$
18: **return** $\mathcal{L}_{DA}$

---

# E    IMPLEMENTATION DETAILS

## E.1    DATASET DETAILS

We provide a brief overview of the datasets used in our evaluation.

- **PACS** (Li et al., 2017): 9,991 images, 7 classes, 4 domains (Photo, Art painting, Cartoon, Sketch).
- **VLCS** (Fang et al., 2013): 10,729 images, 5 classes, 4 domains (PASCAL VOC 2007, LabelMe, Caltech-101, SUN09).
- **OfficeHome** (Venkateswara et al., 2017): 15,588 images, 65 classes, 4 domains (Artistic, Clipart, Product, Real-World).
- **TerraIncognita** (Beery et al., 2018): 24,788 images, 10 classes, 4 camera trap locations (domains).
- **DomainNet** (Peng et al., 2019): Approx. 0.6 million images, 345 classes, 6 domains.

**MDLT Datasets.** The MDLT versions (PACS-MLT, VLCS-MLT, OfficeHome-MLT, TerraIncognita-MLT, and DomainNet-MLT) are from Yang et al. (2022). Crucially, the imbalance factors and the identity of majority/minority classes intentionally diverge across different domains, simulating realistic compound shifts.

## E.2    TRAINING PROTOCOL AND HYPERPARAMETERS

**Architecture and Initialization.** We utilize a standard ResNet-50 architecture. The feature extractor $f_\theta$ is initialized with weights pre-trained on ImageNet-1K. The classifier head $g_\phi$ is initialized randomly.

**General Setup and Optimization.** We adhere to the standard training protocols of the DomainBed framework. Models are typically trained for 5,000 iterations (15,000 for DomainNet-MLT, and DomainNet126-MLT). We use the Adam optimizer (Kingma & Ba, 2015) with a weight decay typically set to $5 \times 10^{-4}$. The batch size is set to 32 in DomainNet, and 24 in others, following Yang et al. (2022).

**Data Augmentation.** We employ standard data augmentations: images are resized and cropped to $224 \times 224$ using Random Resized Crop, followed by Random Horizontal Flip, and Color Jitter (brightness, contrast, saturation=0.4; hue=0.1).

**Hyperparameters and Model Selection.** We strictly adhered to the standardized DomainBed protocol (Gulrajani & Lopez-Paz, 2021) for hyperparameter tuning and model selection. We employed the "Training-domain validation" approach, where models are evaluated on validation sets composed of held-out samples from the training domains.

For all methods (including baselines and RC-Align), hyperparameters were selected by conducting a randomized search (typically 20-50 trials per algorithm) over a predefined joint distribution of hyperparameters, as specified by the DomainBed methodology. The configuration yielding the highest accuracy on the training-domain validation set was selected. This ensures a fair comparison across all methods.

- **Learning Rates:** The base learning rate $\eta$ (used for the outer loop update) and the inner loop learning rate $\alpha$ (used in the adaptation step, Eq. (10)) were searched within the standard DomainBed range (e.g., log-uniform sampling between $10^{-5}$ and $10^{-3}$). When using the FOMAML approximation, $\alpha$ is often searched jointly with or set equal to $\eta$.
- **RC-Align Specific:** The alignment weight $\lambda_{\text{DA}}$ was selected via a search over $\{0.1, 0.5, 1.0, 2.0\}$. The Manifold Mixup parameter $\alpha_{\text{mixup}}$ (used for the Beta distribution sampling) was searched within the standard DomainBed mixup space (e.g., $[0.1, 1.0]$).

We report the average accuracy over three independent runs with random seed.

**Experimental Environment.** The experiments were implemented using PyTorch (v1.10+) and executed on a system equipped with NVIDIA RTX 3090 GPUs (24GB VRAM).

## F  BASELINES FOR MDLT EXPERIMENTS

**Source of baselines.** Unless otherwise noted, all MDLT baselines and their hyperparameters follow the public setup curated in BoDA (Yang et al., 2022). We adopted the same five MDLT benchmarks and the standard training protocol so that numbers are directly comparable.

**Vanilla / Empirical Risk Minimization. ERM** is trained by minimizing the average cross-entropy over pooled domains. We report ERM as implemented in DomainBed (Gulrajani & Lopez-Paz, 2021).

**Domain invariance. IRM** (Arjovsky et al., 2019) learns representations that admit an optimal invariant classifier across environments. **CORAL** aligns second-order feature statistics across domains (Sun & Saenko, 2016). **MMD** minimizes the maximum mean discrepancy between domain feature distributions (Li et al., 2018b). **DANN** (Ganin et al., 2016) uses adversarial training to make features domain-indistinguishable; **CDANN** conditions the discriminator on class predictions to better preserve label information (Li et al., 2018c).

**Robust optimization. GroupDRO** optimizes worst-group risk under group shift to improve worst-domain accuracy (Sagawa* et al., 2020).

**Meta-learning and multi-tasking. MLDG** meta-learns for cross-domain generalization via simulated train/test domain splits (Li et al., 2018a). **MTL** (marginal transfer learning) augments inputs with domain marginals and learns a domain-aware classifier (Blanchard et al., 2021). **Fish** maximizes inter-domain gradient alignment to encourage updates that help across domains (Shi et al., 2021).

**Augmentation and style robustness. Mixup** performs convex interpolations of inputs and labels, improving regularization under compound shifts (Zhang et al., 2018). **SagNet** reduces style bias via style/content disentanglement and adversarial training (Nam et al., 2021).

**Long-tailed (LT/MDLT) baselines. Focal** (Lin et al., 2017) reweights hard examples. **CBLoss** uses the effective number of samples as class weights (Cui et al., 2019). **LDAM** enforces label-distribution-aware margins (Cao et al., 2019). **Balanced Softmax** corrects the softmax gradient bias under label shift (Ren et al., 2020). **CRT (cRT)** decouples representation learning and re-trains a balanced classifier (Kang et al., 2020). **SSP** denotes self-supervised pretraining on the same data followed by supervised fine-tuning, which is known to boost long-tailed recognition (Yang & Xu, 2020).

**BoDA family. BoDA**, **BoDA_r**, **BoDA-M**, and **BoDA_r,c** share the same objective family that balances cross-domain alignment and within-domain calibration under class imbalance, with suffixes indicating coupled/decoupled classifier training and margin variants (Yang et al., 2022).

**Our method. RC-Align (Ours)** is evaluated under the identical data splits, model backbones, and early-stopping rules as above to ensure fair comparison.

## G  DETAILED RESULTS FOR DOMAIN GENERALIZATION

This section provides the detailed results for domain generalization benchmark.

Table 4: Domain generalization accuracy (%) on the PACS dataset. Results are sourced from their original papers unless otherwise noted. † Gulrajani & Lopez-Paz (2021); ‡ Cha et al. (2021); § Jia & Zhang (2024). The best average performance is highlighted in **bold**.

| Algorithm | A | C | P | S | Avg |
|---|---|---|---|---|---|
| ERM† (Vapnik, 1998) | 84.7 | 80.8 | 97.2 | 79.3 | 85.5 |
| IRM† (Arjovsky et al., 2019) | 84.8 | 76.4 | 96.7 | 76.1 | 83.5 |
| GroupDRO† (Sagawa* et al., 2020) | 83.5 | 79.1 | 96.7 | 78.3 | 84.4 |
| Mixup† (Zhang et al., 2018) | 86.1 | 78.9 | 97.6 | 75.8 | 84.6 |
| MLDG† (Li et al., 2018a) | 85.5 | 80.1 | 97.4 | 76.6 | 84.9 |
| CORAL† (Sun & Saenko, 2016) | 88.3 | 80.0 | 97.5 | 78.8 | 86.2 |
| MMD† (Li et al., 2018b) | 86.1 | 79.4 | 96.6 | 76.5 | 84.6 |
| DANN† (Ganin et al., 2016) | 86.4 | 77.4 | 97.3 | 73.5 | 83.7 |
| CDANN† (Li et al., 2018c) | 84.6 | 75.5 | 96.8 | 73.5 | 82.6 |
| MTL† (Blanchard et al., 2021) | 87.5 | 77.1 | 96.4 | 77.3 | 84.6 |
| SagNet† (Nam et al., 2021) | 87.4 | 80.7 | 97.1 | 80.0 | 86.3 |
| ARM† (Zhang et al., 2020) | 86.8 | 76.8 | 97.4 | 79.3 | 85.1 |
| VREx† (Krueger et al., 2020) | 86.0 | 79.1 | 96.9 | 77.7 | 84.9 |
| RSC† (Huang et al., 2020) | 85.4 | 79.7 | 97.6 | 78.2 | 85.2 |
| MetaReg§ (Balaji et al., 2018) | - | - | - | - | 83.6 |
| MLIR§ (Jia & Zhang, 2024) | - | - | - | - | 86.8 |
| Mixstyle‡ (Zhou et al., 2021) | 86.8 | 79.0 | 96.6 | 78.5 | 85.2 |
| SWAD‡ (Cha et al., 2021) | 89.3 | 83.4 | 97.3 | 82.5 | 88.1 |
| PCL Yao et al. (2022) | 90.2 | **83.9** | 98.1 | 82.6 | 88.7 |
| BoDA (Yang et al., 2022) | 88.2 | 81.7 | 97.8 | 80.2 | 86.9 |
| SAGM (Wang et al., 2023) | 87.4 | 80.2 | 98.0 | 80.8 | 86.6 |
| iDAG (Huang et al., 2023) | **90.8** | 83.7 | 98.0 | **82.7** | **88.8** |
| GMDG (Tan et al., 2024) | 84.7 | 81.7 | 97.5 | 80.5 | 85.6 |
| Arith (Wang et al., 2025) | 85.9 | 81.3 | 97.1 | 81.8 | 86.5 |
| **Ours** | 89.8±0.6 | 79.9±0.2 | **98.8** ±0.2 | 81.3±1.0 | 87.5 |

Table 5: Domain generalization accuracy (%) on the VLCS dataset. Results are sourced from their original papers unless otherwise noted. † Gulrajani & Lopez-Paz (2021); ‡ Cha et al. (2021); § Jia & Zhang (2024). The best average performance is highlighted in **bold**.

| Algorithm | C | L | S | V | Avg |
|---|---|---|---|---|---|
| ERM† (Vapnik, 1998) | 97.7 | 64.3 | 73.4 | 74.6 | 77.5 |
| IRM† (Arjovsky et al., 2019) | 98.6 | 64.9 | 73.4 | 77.3 | 78.5 |
| GroupDRO† (Sagawa* et al., 2020) | 97.3 | 63.4 | 69.5 | 76.7 | 76.7 |
| Mixup† (Zhang et al., 2018) | 98.3 | 64.8 | 72.1 | 74.3 | 77.4 |
| MLDG† (Li et al., 2018a) | 97.4 | 65.2 | 71.0 | 75.3 | 77.2 |
| CORAL† (Sun & Saenko, 2016) | 98.3 | 66.1 | 73.4 | 77.5 | 78.8 |
| MMD† (Li et al., 2018b) | 97.7 | 64.0 | 72.8 | 75.3 | 77.5 |
| DANN† (Ganin et al., 2016) | **99.0** | 65.1 | 73.1 | 77.2 | 78.6 |
| CDANN† (Li et al., 2018c) | 97.1 | 65.1 | 70.7 | 77.1 | 77.5 |
| MTL† (Blanchard et al., 2021) | 97.8 | 64.3 | 71.5 | 75.3 | 77.2 |
| SagNet† (Nam et al., 2021) | 97.9 | 64.5 | 71.4 | 77.5 | 77.8 |
| ARM† (Zhang et al., 2020) | 98.7 | 63.6 | 71.3 | 76.7 | 77.6 |
| VREx† (Krueger et al., 2020) | 98.4 | 64.4 | 74.1 | 76.2 | 78.3 |
| RSC† (Huang et al., 2020) | 97.9 | 62.5 | 72.3 | 75.6 | 77.1 |
| MetaReg§ (Balaji et al., 2018) | - | - | - | - | 76.7 |
| MLIR§ (Jia & Zhang, 2024) | - | - | - | - | 80.7 |
| Mixstyle‡ (Zhou et al., 2021) | 98.6 | 64.5 | 72.6 | 75.7 | 77.9 |
| SWAD‡ (Cha et al., 2021) | 98.8 | 63.3 | 75.3 | 79.2 | 79.1 |
| PCL Yao et al. (2022) | **99.0** | 63.6 | 73.8 | 75.6 | 78.0 |
| BoDA (Yang et al., 2022) | 98.1 | 64.5 | 74.3 | 78.0 | 78.5 |
| SAGM (Wang et al., 2023) | **99.0** | 65.2 | 75.1 | **80.7** | 80.0 |
| iDAG (Huang et al., 2023) | 98.1 | 62.7 | 69.9 | 77.1 | 76.9 |
| GMDG (Tan et al., 2024) | 98.3 | 65.9 | 73.4 | 79.3 | 79.2 |
| Arith (Wang et al., 2025) | 98.7 | 64.6 | 76.3 | 77.8 | 79.4 |
| **Ours** | 97.5±0.4 | **68.5**±0.4 | **78.1**±0.8 | 79.9±0.9 | **81.0** |

Table 6: Domain generalization accuracy (%) on the OfficeHome dataset. Results are sourced from their original papers unless otherwise noted. † Gulrajani & Lopez-Paz (2021); ‡ Cha et al. (2021); § Jia & Zhang (2024). The best average performance is highlighted in **bold**.

| Algorithm | A | C | P | R | Avg |
|---|---|---|---|---|---|
| ERM[†] (Vapnik, 1998) | 61.3 | 52.4 | 75.8 | 76.6 | 66.5 |
| IRM[†] (Arjovsky et al., 2019) | 58.9 | 52.2 | 72.1 | 74.0 | 64.3 |
| GroupDRO[†] (Sagawa* et al., 2020) | 60.4 | 52.7 | 75.0 | 76.0 | 66.0 |
| Mixup[†] (Zhang et al., 2018) | 62.4 | 54.8 | 76.9 | 78.3 | 68.1 |
| MLDG[†] (Li et al., 2018a) | 61.5 | 53.2 | 75.0 | 77.5 | 66.8 |
| CORAL[†] (Sun & Saenko, 2016) | 65.3 | 54.4 | 76.5 | 78.4 | 68.7 |
| MMD[†] (Li et al., 2018b) | 60.4 | 53.3 | 74.3 | 77.4 | 66.3 |
| DANN[†] (Ganin et al., 2016) | 59.9 | 53.0 | 73.6 | 76.9 | 65.9 |
| CDANN[†] (Li et al., 2018c) | 61.5 | 50.4 | 74.4 | 76.6 | 65.8 |
| MTL[†] (Blanchard et al., 2021) | 61.5 | 52.4 | 74.9 | 76.8 | 66.4 |
| SagNet[†] (Nam et al., 2021) | 63.4 | 54.8 | 75.8 | 78.3 | 68.1 |
| ARM[†] (Zhang et al., 2020) | 58.9 | 51.0 | 74.1 | 75.2 | 64.8 |
| VREx[†] (Krueger et al., 2020) | 60.7 | 53.0 | 75.3 | 76.6 | 66.4 |
| RSC[†] (Huang et al., 2020) | 60.7 | 51.4 | 74.8 | 75.1 | 65.5 |
| MetaReg[§] (Balaji et al., 2018) | - | - | - | - | 67.6 |
| MLIR[§] (Jia & Zhang, 2024) | - | - | - | - | 69.8 |
| Mixstyle[‡] (Zhou et al., 2021) | 51.1 | 53.2 | 68.2 | 69.2 | 60.4 |
| SWAD[‡] (Cha et al., 2021) | 66.1 | 57.7 | 78.4 | 80.2 | 70.6 |
| PCL Yao et al. (2022) | 67.3 | 59.9 | 78.7 | 80.7 | 71.6 |
| BoDA (Yang et al., 2022) | 65.4 | 55.4 | 77.1 | 79.5 | 69.3 |
| SAGM (Wang et al., 2023) | 65.4 | 57.0 | 78.0 | 80.0 | 70.1 |
| iDAG (Huang et al., 2023) | 68.2 | **57.9** | **79.7** | **81.4** | **71.8** |
| GMDG (Tan et al., 2024) | - | - | - | - | 70.7 |
| Arith (Wang et al., 2025) | 64.7 | 56.3 | 77.5 | 79.2 | 69.4 |
| **Ours** | **69.4**±0.1 | 54.4±0.4 | 79.0±0.2 | 80.9±0.2 | 70.9 |

Table 7: Domain generalization accuracy (%) on the TerraIncognita dataset. Results are sourced from their original papers unless otherwise noted. † Gulrajani & Lopez-Paz (2021); ‡ Cha et al. (2021); § Jia & Zhang (2024). The best average performance is highlighted in **bold**.

| Algorithm | L100 | L38 | L43 | L46 | Avg |
|---|---|---|---|---|---|
| ERM† (Vapnik, 1998) | 49.8 | 42.1 | 56.9 | 35.7 | 46.1 |
| IRM† (Arjovsky et al., 2019) | 54.6 | 39.8 | 56.2 | 39.6 | 47.6 |
| GroupDRO† (Sagawa* et al., 2020) | 41.2 | 38.6 | 56.7 | 36.4 | 43.2 |
| Mixup† (Zhang et al., 2018) | 59.6 | 42.2 | 55.9 | 33.9 | 47.9 |
| MLDG† (Li et al., 2018a) | 54.2 | 44.3 | 55.6 | 36.9 | 47.7 |
| CORAL† (Sun & Saenko, 2016) | 51.6 | 42.2 | 57.0 | 39.8 | 47.6 |
| MMD† (Li et al., 2018b) | 41.9 | 34.8 | 57.0 | 35.2 | 42.2 |
| DANN† (Ganin et al., 2016) | 51.1 | 40.6 | 57.4 | 37.7 | 46.7 |
| CDANN† (Li et al., 2018c) | 47.0 | 41.3 | 54.9 | 39.8 | 45.8 |
| MTL† (Blanchard et al., 2021) | 49.3 | 39.6 | 55.6 | 37.8 | 45.6 |
| SagNet† (Nam et al., 2021) | 53.0 | 43.0 | 57.9 | 40.4 | 48.6 |
| ARM† (Zhang et al., 2020) | 49.3 | 38.3 | 55.8 | 38.7 | 45.5 |
| VREx† (Krueger et al., 2020) | 48.2 | 41.7 | 56.8 | 38.7 | 46.4 |
| RSC† (Huang et al., 2020) | 50.2 | 39.2 | 56.3 | 40.8 | 46.6 |
| MetaReg§ (Balaji et al., 2018) | - | - | - | - | 48.2 |
| MLIR§ (Jia & Zhang, 2024) | - | - | - | - | 51.0 |
| Mixstyle‡ (Zhou et al., 2021) | 54.3 | 34.1 | 55.9 | 31.7 | 44.0 |
| SWAD‡ (Cha et al., 2021) | 55.4 | 44.9 | 59.7 | 39.9 | 50.0 |
| PCL Yao et al. (2022) | 58.7 | 46.3 | **60.0** | **43.6** | 52.1 |
| BoDA (Yang et al., 2022) | 54.0 | 46.5 | 59.5 | 41.0 | 50.2 |
| SAGM (Wang et al., 2023) | 54.8 | 41.4 | 57.7 | 41.3 | 48.8 |
| iDAG (Huang et al., 2023) | 58.7 | 35.1 | 57.5 | 33.0 | 46.1 |
| GMDG (Tan et al., 2024) | 59.8 | 45.3 | 57.1 | 38.2 | 50.1 |
| Arith (Wang et al., 2025) | - | - | - | - | 48.1 |
| **Ours** | **63.2**±0.5 | **50.7**±0.2 | 54.9±0.3 | 41.5±0.5 | **52.6** |

# H    DETAILED RESULTS FOR MDLT BENCHMARKS

This section provides the detailed results for MDLT benchmarks. In this section, we add experimental results from DomainNet126 (Saito et al., 2019).

**DomainNet126-MLT.** We construct DomainNet126-MLT based on the DomainNet126 dataset (Saito et al., 2019), a curated subset of the large-scale DomainNet benchmark (Peng et al., 2019), to alleviate issues arising from noisy labels in specific domains and classes. DomainNet126-MLT contains 122,505 images from 4 domains (*clipart, painting, real, sketch*) with 126 classes. We partitioned the dataset into training, validation, and test subsets. The validation and test sets were constructed as balanced subsets, comprising approximately 5% (8,132 images) and 10% (16,377 images) of the full dataset, respectively, while the remaining 97,995 images were allocated for training. The procedure for constructing the validation and test sets follows the protocol outlined in (Yang et al., 2022).

DomainNet126-MLT exhibits significant class imbalance both within and across domains. For example, in the *real* domain, class frequencies range from 134 to 802 images per class (mean $\approx$ 553), whereas in *painting* the distribution is much sparser and skewed, ranging from only 1 to 249 images per class (mean $\approx$ 81). This imbalance makes DomainNet126-MLT a challenging and realistic benchmark for multi-domain long-tailed recognition. It should be noted that Table 13 excludes SSP (Yang & Xu, 2020), as the corresponding code and model have not been fully made available. Figure 2 shows the label distributions across domains in the DomainNet126-MLT.

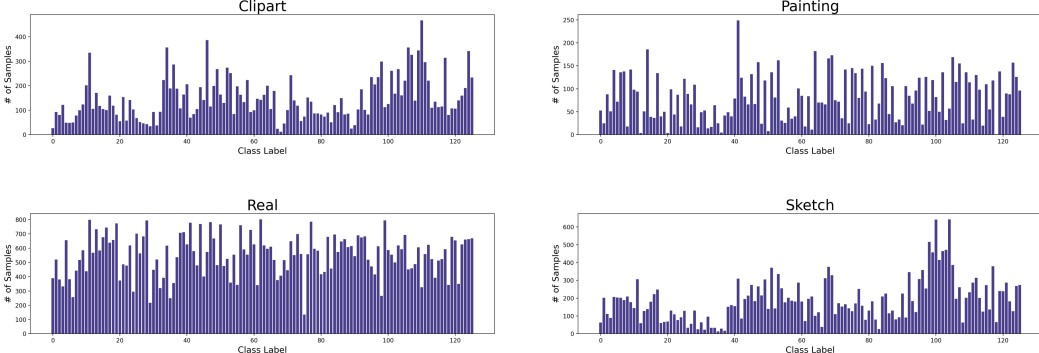

Figure 2: Class distributions across the four domains in DomainNet126-MLT. The dataset exhibits a clear long-tailed distribution within and across domains.

Table 8: Accuracy (%) under the multi-domain long-tailed (MDLT) setting on **PACS-MLT**. Results are sourced from their original papers, and are reported by domain and by shot.

| Algorithm | Accuracy (by domain) | | | | | | Accuracy (by shot) | | | |
| --- | --- | --- | --- | --- | --- | --- | --- | --- | --- | --- |
| | A | C | P | S | Average | Worst | Many | Medium | Few | Zero |
| ERM (Vapnik, 1998) | 96.8±0.1 | 97.0±0.3 | 98.9±0.3 | 95.8±0.2 | 97.1±0.1 | 95.8±0.2 | 97.1±0.0 | 97.0±0.0 | 98.0±0.9 | - |
| IRM (Arjovsky et al., 2019) | 96.8±0.1 | 96.3±0.7 | 98.7±0.2 | 95.2±0.4 | 96.7±0.2 | 95.2±0.4 | 96.8±0.2 | 96.7±0.7 | 94.7±1.4 | - |
| GroupDRO (Sagawa* et al., 2020) | 96.9±0.2 | 97.0±0.4 | 99.0±0.1 | 95.3±0.4 | 97.0±0.1 | 95.3±0.4 | 97.3±0.1 | 95.3±1.2 | 94.7±3.6 | - |
| Mixup (Zhang et al., 2018) | 96.5±0.3 | 96.9±0.7 | 98.5±0.2 | 95.1±0.2 | 96.7±0.2 | 95.1±0.2 | 97.0±0.1 | 96.7±0.3 | 91.3±2.7 | - |
| MLDG (Li et al., 2018a) | 96.6±0.2 | 97.2±0.3 | 98.5±0.1 | 94.1±0.3 | 96.6±0.1 | 94.1±0.3 | 96.8±0.1 | 96.3±0.7 | 92.7±0.5 | - |
| CORAL (Sun & Saenko, 2016) | 96.9±0.4 | 97.0±0.5 | 98.3±0.3 | 94.3±0.7 | 96.6±0.5 | 94.3±0.7 | 96.6±0.5 | 97.0±0.8 | 94.7±0.5 | - |
| MMD (Li et al., 2018b) | 96.8±0.2 | 97.1±0.4 | 97.4±0.3 | 96.3±0.3 | 96.9±0.1 | 96.2±0.2 | 96.9±0.2 | 97.0±0.0 | 96.7±0.5 | - |
| DANN (Ganin et al., 2016) | 95.7±0.3 | 97.2±0.4 | 98.9±0.1 | 94.3±0.1 | 96.5±0.0 | 94.3±0.1 | 96.5±0.1 | 98.0±0.0 | 94.7±2.4 | - |
| CDANN (Li et al., 2018c) | 95.5±0.5 | 96.7±0.2 | 97.2±0.3 | 94.9±0.5 | 96.1±0.1 | 94.5±0.2 | 96.1±0.1 | 96.3±0.5 | 94.0±0.9 | - |
| MTL (Blanchard et al., 2021) | 96.3±0.4 | 97.9±0.3 | 98.2±0.3 | 94.6±0.7 | 96.7±0.2 | 94.5±0.6 | 96.8±0.1 | 95.3±1.7 | 97.3±1.1 | - |
| SagNet (Nam et al., 2021) | 97.0±0.2 | 97.8±0.4 | 98.9±0.1 | 95.2±0.3 | 97.2±0.1 | 95.2±0.3 | 97.4±0.1 | 96.7±0.5 | 95.3±0.5 | - |
| Fish (Shi et al., 2021) | 95.5±0.2 | 97.9±0.4 | 98.2±0.3 | 95.9±0.5 | 96.9±0.2 | 95.2±0.2 | 97.0±0.1 | 97.0±0.5 | 94.7±1.1 | - |
| Focal (Lin et al., 2017) | 96.6±0.4 | 96.6±0.8 | 98.1±0.2 | 94.6±0.7 | 96.5±0.2 | 94.6±0.7 | 96.6±0.1 | 95.0±1.7 | 96.7±0.5 | - |
| CBLoss (Cui et al., 2019) | 97.3±0.1 | 97.4±0.5 | 97.8±0.6 | 95.1±0.4 | 96.9±0.1 | 95.1±0.4 | 96.8±0.2 | 97.0±1.2 | **100.0**±0.0 | - |
| LDAM (Cao et al., 2019) | 96.9±0.1 | 96.6±0.6 | 97.9±0.1 | 94.7±0.2 | 96.5±0.2 | 94.7±0.2 | 96.6±0.1 | 95.7±1.4 | 96.0±0.0 | - |
| BSoftmax (Ren et al., 2020) | 96.0±0.5 | 96.9±0.6 | 98.8±0.6 | 95.9±0.1 | 96.9±0.3 | 95.6±0.3 | 96.6±0.4 | 98.7±0.7 | 99.3±0.5 | - |
| SSP (Yang & Xu, 2020) | 96.2±0.5 | 96.8±0.2 | 98.9±0.1 | 95.7±0.3 | 96.9±0.2 | 95.4±0.4 | 96.7±0.2 | 98.3±0.5 | 98.0±0.9 | - |
| CRT (Kang et al., 2020) | 95.3±0.2 | 96.7±0.1 | 98.5±0.1 | 94.9±0.1 | 96.3±0.1 | 94.9±0.1 | 96.3±0.1 | 97.3±0.3 | 94.0±0.9 | - |
| BoDAr (Yang et al., 2022) | 96.9±0.4 | 97.4±0.2 | 98.6±0.2 | 95.1±0.4 | 97.0±0.1 | 95.1±0.4 | 97.0±0.1 | 96.3±0.5 | 98.0±0.9 | - |
| BoDA-Mr (Yang et al., 2022) | 96.6±0.2 | 98.0±0.2 | 99.1±0.2 | 94.9±0.1 | 97.1±0.1 | 94.9±0.1 | 97.3±0.1 | 96.3±0.5 | 96.0±0.0 | - |
| BoDAr,c (Yang et al., 2022) | 96.3±0.1 | 97.4±0.5 | 99.4±0.3 | 95.7±0.3 | 97.2±0.1 | 95.7±0.3 | 97.4±0.1 | 97.0±0.0 | 94.7±1.1 | - |
| BoDA-Mr,c (Yang et al., 2022) | 96.3±0.4 | 97.7±0.2 | 98.1±0.4 | 96.4±0.2 | 97.1±0.2 | 96.3±0.1 | 97.1±0.0 | 97.0±0.8 | 96.0±0.0 | - |
| **Ours** | **98.3**±0.3 | **98.2**±0.1 | **99.6**±0.3 | **97.4**±0.5 | **98.4**±0.3 | **97.4**±0.5 | **98.3**±0.3 | **99.7**±0.3 | 98.7±0.7 | - |

Table 9: Accuracy (%) under the multi-domain long-tailed (MDLT) setting on **VLCS-MLT**. Results are sourced from their original papers, and are reported by domain and by shot.

| Algorithm | Accuracy (by domain) | | | | | | Accuracy (by shot) | | | |
| --- | --- | --- | --- | --- | --- | --- | --- | --- | --- | --- |
| | C | L | S | V | Average | Worst | Many | Medium | Few | Zero |
| ERM (Vapnik, 1998) | 99.3±0.3 | 53.6±1.1 | 65.9±1.2 | 86.4±0.7 | 76.3±0.4 | 53.6±1.1 | 84.6±0.5 | 76.6±0.4 | - | 32.9±0.4 |
| IRM (Arjovsky et al., 2019) | 99.1±0.4 | 52.3±0.7 | 68.8±1.4 | 86.0±0.3 | 76.5±0.2 | 52.3±0.7 | 85.3±0.6 | 75.5±1.0 | - | 33.5±1.0 |
| GroupDRO (Sagawa* et al., 2020) | 98.7±0.3 | 54.1±1.3 | 67.5±1.5 | 86.7±0.3 | 76.7±0.4 | 54.1±1.3 | 85.3±0.9 | 76.2±1.0 | - | 34.5±2.0 |
| Mixup (Zhang et al., 2018) | 99.3±0.3 | 52.7±1.3 | 66.1±0.0 | 85.3±1.1 | 75.9±0.1 | 52.7±1.3 | 84.4±0.2 | 77.1±0.6 | - | 29.2±1.4 |
| MLDG (Li et al., 2018a) | 99.3±0.3 | 53.6±0.5 | 68.3±0.4 | 86.4±0.5 | 76.9±0.2 | 53.6±0.5 | 84.9±0.3 | 77.5±1.0 | - | 34.4±0.9 |
| CORAL (Sun & Saenko, 2016) | 99.3±0.3 | 51.6±0.7 | 67.5±1.8 | 85.3±0.9 | 75.9±0.5 | 51.6±0.7 | 84.5±0.8 | 75.5±0.5 | - | 34.5±0.8 |
| MMD (Li et al., 2018b) | 99.6±0.2 | 53.4±0.3 | 65.6±0.8 | 86.7±1.1 | 76.3±0.6 | 53.4±0.3 | 84.5±0.8 | 77.1±0.5 | - | 32.7±0.3 |
| DANN (Ganin et al., 2016) | 99.6±0.2 | 54.1±0.3 | 69.9±0.2 | 86.7±0.0 | 77.5±0.1 | 54.1±0.3 | 85.9±0.5 | 76.0±0.4 | - | 38.0±2.3 |
| CDANN (Li et al., 2018c) | 99.6±0.4 | 53.6±0.4 | 67.5±0.6 | 85.8±0.8 | 76.6±0.4 | 53.6±0.4 | 84.4±0.7 | 77.3±0.8 | - | 35.0±0.8 |
| MTL (Blanchard et al., 2021) | 99.1±0.2 | 52.9±0.5 | 66.8±0.2 | 86.7±0.6 | 76.3±0.3 | 52.9±0.5 | 84.8±0.9 | 76.2±0.6 | - | 33.3±1.4 |
| SagNet (Nam et al., 2021) | 99.6±0.4 | 52.3±0.2 | 67.2±0.2 | 86.2±1.0 | 76.3±0.2 | 52.3±0.2 | 85.3±0.3 | 75.1±0.2 | - | 32.9±0.3 |
| Fish (Shi et al., 2021) | 98.7±0.3 | 54.3±0.4 | 69.4±0.8 | 87.6±0.4 | 77.5±0.3 | 54.3±0.4 | 86.2±0.5 | 76.0±0.4 | - | 35.6±2.2 |
| Focal (Lin et al., 2017) | 99.1±0.4 | 52.3±0.2 | 66.1±0.8 | 84.9±0.2 | 75.6±0.4 | 52.3±0.2 | 84.0±0.2 | 75.5±0.6 | - | 32.7±0.9 |
| CBLoss (Cui et al., 2019) | 99.1±0.2 | 52.5±0.5 | 68.5±1.0 | 87.1±1.0 | 76.8±0.3 | 52.5±0.5 | 84.8±0.7 | 77.5±1.4 | - | 33.2±1.6 |
| LDAM (Cao et al., 2019) | 98.9±0.2 | 52.9±0.2 | 69.4±1.4 | 88.0±1.3 | 77.5±0.1 | 52.9±0.2 | 86.5±0.4 | 75.5±0.5 | - | 35.2±0.6 |
| BSoftmax (Ren et al., 2020) | 99.3±0.3 | 52.9±0.9 | 68.0±0.2 | 86.7±0.8 | 76.7±0.5 | 52.9±0.9 | 84.4±0.9 | 78.2±0.6 | - | 34.3±0.9 |
| SSP (Yang & Xu, 2020) | 99.1±0.2 | 52.3±1.0 | 68.0±0.2 | 85.1±0.4 | 76.1±0.3 | 52.3±1.0 | 83.8±0.3 | 76.0±1.2 | - | 37.1±0.7 |
| CRT (Kang et al., 2020) | 99.6±0.3 | 51.4±0.3 | 66.9±0.8 | 86.9±0.4 | 76.3±0.2 | 51.4±0.3 | 84.5±0.1 | 77.3±0.0 | - | 31.7±1.0 |
| BoDAr (Yang et al., 2022) | 99.3±0.3 | 51.4±0.3 | 70.2±0.4 | 86.7±0.3 | 76.9±0.5 | 51.4±0.3 | 85.3±0.3 | 77.3±0.2 | - | 33.3±0.5 |
| BoDA-Mr (Yang et al., 2022) | **100.0**±0.0 | 53.4±0.3 | 68.8±0.8 | 87.5±0.3 | 77.5±0.3 | 53.4±0.3 | 85.8±0.2 | 77.3±0.2 | - | 35.7±0.7 |
| BoDAr,c (Yang et al., 2022) | 99.3±0.3 | 53.4±0.3 | 68.5±0.4 | 88.0±0.4 | 77.3±0.2 | 53.4±0.3 | 85.3±0.3 | 78.0±0.2 | - | 38.6±0.7 |
| BoDA-Mr,c (Yang et al., 2022) | **100.0**±0.0 | 55.4±0.5 | **72.6**±0.3 | 84.7±0.5 | 78.2±0.4 | 55.4±0.5 | 85.3±0.3 | 79.3±0.6 | - | **43.3**±1.1 |
| **Ours** | 99.8±0.2 | **57.4**±0.8 | 70.4±1.0 | **88.7**±0.8 | **79.1**±0.2 | **57.4**±0.8 | **86.9**±0.9 | **79.5**±1.2 | - | 39.4±1.6 |

Table 10: Accuracy (%) under the multi-domain long-tailed (MDLT) setting on **OfficeHome**. Results are sourced from their original papers, and are reported by domain and by shot.

| | Accuracy (by domain) | | | | | | Accuracy (by shot) | | | |
|---|---|---|---|---|---|---|---|---|---|---|
| Algorithm | A | C | P | R | Average | Worst | Many | Medium | Few | Zero |
| ERM (Vapnik, 1998) | 71.3±0.1 | 78.4±0.2 | 89.6±0.3 | 83.3±0.2 | 80.7±0.0 | 71.3±0.1 | 87.8±0.2 | 81.0±0.2 | 63.1±0.1 | 63.3±7.2 |
| IRM (Arjovsky et al., 2019) | 70.7±0.2 | 78.5±0.8 | 89.4±0.5 | 83.8±0.6 | 80.6±0.4 | 70.7±0.2 | 87.6±0.4 | 81.5±0.4 | 61.1±0.9 | 56.7±1.4 |
| GroupDRO (Sagawa* et al., 2020) | 68.7±0.9 | 79.0±0.2 | 89.4±0.4 | 83.3±0.5 | 80.1±0.3 | 68.7±0.9 | 88.1±0.2 | 80.8±0.4 | 59.8±1.2 | 51.7±3.6 |
| Mixup (Zhang et al., 2018) | 72.3±0.6 | 79.1±0.4 | 89.7±0.1 | 83.9±0.2 | 81.2±0.2 | 72.3±0.6 | 87.9±0.4 | 81.8±0.1 | 64.1±0.4 | 60.0±4.1 |
| MLDG (Li et al., 2018a) | 70.2±0.6 | 78.2±0.5 | 89.4±0.4 | 83.7±0.3 | 80.4±0.2 | 70.2±0.6 | 87.1±0.1 | 81.3±0.3 | 61.3±1.0 | 61.7±1.4 |
| CORAL (Sun & Saenko, 2016) | 72.7±0.6 | 80.9±0.3 | 89.9±0.2 | 84.2±0.4 | 81.9±0.1 | 72.7±0.6 | 87.9±0.1 | 83.0±0.1 | 63.5±0.7 | 65.0±2.4 |
| MMD (Li et al., 2018b) | 67.7±0.8 | 77.8±0.2 | 87.4±0.5 | 80.6±0.4 | 78.4±0.4 | 67.7±0.8 | 85.2±0.2 | 79.4±0.7 | 58.8±0.4 | 56.7±3.6 |
| DANN (Ganin et al., 2016) | 70.2±0.9 | 77.3±0.3 | 87.3±0.5 | 82.1±0.4 | 79.2±0.2 | 70.2±0.9 | 86.2±0.1 | 80.0±0.1 | 60.3±1.1 | 61.7±5.9 |
| CDANN (Li et al., 2018c) | 69.4±0.3 | 77.2±0.3 | 87.7±0.2 | 81.5±0.3 | 79.0±0.2 | 69.4±0.3 | 86.4±0.6 | 79.8±0.1 | 58.9±0.8 | 50.0±4.7 |
| MTL (Blanchard et al., 2021) | 69.8±0.6 | 77.6±0.3 | 87.9±0.1 | 82.4±0.3 | 79.5±0.2 | 69.8±0.6 | 87.3±0.3 | 79.8±0.2 | 61.1±0.2 | 51.7±2.7 |
| SagNet (Nam et al., 2021) | 70.5±0.5 | 79.6±0.5 | 89.3±0.4 | 83.9±0.1 | 80.9±0.1 | 70.5±0.5 | 87.8±0.4 | 81.9±0.1 | 61.2±0.9 | 56.7±3.6 |
| Fish (Shi et al., 2021) | 71.3±0.1 | 79.1±0.1 | 90.2±0.6 | 84.7±0.8 | 81.3±0.3 | 71.3±0.7 | 88.2±0.2 | 81.9±0.3 | 63.2±0.8 | 61.7±1.4 |
| Focal (Lin et al., 2017) | 67.6±0.4 | 76.6±0.8 | 87.1±0.5 | 80.2±0.3 | 77.9±0.0 | 67.6±0.4 | 86.5±0.3 | 78.3±0.1 | 57.4±0.3 | 46.7±3.6 |
| CBLoss (Cui et al., 2019) | 69.5±0.7 | 78.7±0.3 | 88.9±0.4 | 82.2±0.1 | 79.8±0.2 | 69.5±0.7 | 86.6±0.4 | 80.6±0.2 | 61.1±1.4 | 65.0±2.4 |
| LDAM (Cao et al., 2019) | 69.9±0.5 | 78.9±0.4 | 89.4±0.3 | 83.0±0.4 | 80.3±0.2 | 69.9±0.5 | 87.1±0.2 | 81.3±0.3 | 61.1±0.2 | 51.7±2.7 |
| BSoftmax (Ren et al., 2020) | 70.9±0.6 | 78.7±0.2 | 89.0±0.8 | 83.0±0.3 | 80.4±0.2 | 70.9±0.5 | 86.7±0.5 | 81.3±0.3 | 62.4±1.0 | 60.0±4.1 |
| SSP (Yang & Xu, 2020) | 71.1±0.3 | 79.6±0.8 | 89.4±0.3 | 84.2±0.3 | 81.1±0.3 | 71.1±0.3 | 87.3±0.6 | 82.3±0.3 | 61.6±0.7 | 63.3±1.4 |
| CRT (Kang et al., 2020) | 72.5±0.2 | 79.6±0.2 | 88.9±0.1 | 83.6±0.2 | 81.2±0.0 | 72.5±0.2 | 87.7±0.1 | 81.8±0.1 | 64.0±0.1 | 65.0±2.4 |
| BoDAr (Yang et al., 2022) | 71.8±0.1 | 80.3±0.3 | 89.1±0.4 | 84.6±0.2 | 81.5±0.1 | 71.8±0.1 | 87.7±0.2 | 82.6±0.1 | 64.2±0.3 | 63.3±1.4 |
| BoDA-Mr (Yang et al., 2022) | 71.6±0.2 | 80.5±0.3 | 89.2±0.2 | 85.7±0.4 | 81.9±0.2 | 71.6±0.2 | 87.3±0.3 | 83.4±0.2 | 62.3±0.3 | 65.0±2.4 |
| BoDAr,c (Yang et al., 2022) | 72.3±0.3 | 80.8±0.2 | 89.4±0.4 | 86.3±0.3 | 82.3±0.1 | 72.3±0.3 | 87.1±0.2 | 83.9±0.3 | 63.2±0.2 | 65.0±2.4 |
| BoDA-Mr,c (Yang et al., 2022) | 72.3±0.3 | **81.5**±0.4 | 89.5±0.3 | 85.8±0.2 | 82.4±0.2 | 72.3±0.3 | 87.7±0.1 | 83.9±0.6 | 64.2±0.3 | **66.7**±2.7 |
| **Ours** | **75.4**±0.1 | 79.2±0.2 | **92.1**±0.1 | **86.7**±0.3 | **83.3**±0.0 | **75.4**±0.1 | **89.4**±0.2 | **84.1**±0.1 | **67.1**±0.2 | 56.7±3.3 |

Table 11: Accuracy (%) under the multi-domain long-tailed (MDLT) setting on **TerraInc-MLT**. Results are sourced from their original papers, and are reported by domain and by shot.

| | Accuracy (by domain) | | | | | | Accuracy (by shot) | | | |
|---|---|---|---|---|---|---|---|---|---|---|
| Algorithm | L100 | L38 | L43 | L46 | Average | Worst | Many | Medium | Few | Zero |
| ERM (Vapnik, 1998) | 80.3±1.3 | 71.2±0.7 | 82.2±0.3 | 67.4±0.3 | 75.3±0.3 | 67.4±0.3 | 85.6±0.8 | 69.6±3.2 | 66.1±2.4 | 14.4±2.8 |
| IRM (Arjovsky et al., 2019) | 78.2±0.9 | 69.6±2.0 | 81.1±0.7 | 64.3±1.3 | 73.3±0.7 | 64.3±1.3 | 83.5±0.6 | 70.0±1.8 | 58.3±3.4 | 20.1±1.4 |
| GroupDRO (Sagawa* et al., 2020) | 68.3±1.0 | 68.8±1.3 | 82.6±0.2 | 68.1±0.8 | 72.0±0.4 | 66.6±0.2 | 84.7±1.1 | 64.6±4.7 | 38.9±1.2 | 13.5±1.1 |
| Mixup (Zhang et al., 2018) | 75.4±1.4 | 70.2±1.3 | 78.3±0.6 | 60.4±1.1 | 71.1±0.7 | 60.4±1.1 | 83.2±0.7 | 60.0±0.6 | 56.1±3.0 | 12.2±2.1 |
| MLDG (Li et al., 2018a) | 82.3±0.9 | 73.5±2.0 | 83.8±1.4 | 66.9±0.5 | 76.6±0.2 | 66.9±0.5 | 86.1±0.6 | 73.8±3.9 | 70.6±3.7 | 18.8±2.4 |
| CORAL (Sun & Saenko, 2016) | 81.6±1.0 | 72.0±0.6 | 84.2±0.2 | 67.8±0.9 | 76.4±0.5 | 67.8±0.9 | 86.3±0.3 | 77.5±3.1 | 66.1±2.0 | 11.0±1.4 |
| MMD (Li et al., 2018b) | 78.9±0.6 | 68.8±1.0 | 81.9±0.9 | 63.7±1.1 | 73.3±0.4 | 63.7±1.1 | 84.0±0.4 | 67.9±2.7 | 60.6±1.6 | 13.6±2.6 |
| DANN (Ganin et al., 2016) | 74.1±0.8 | 63.1±1.9 | 75.9±0.2 | 61.5±0.9 | 68.7±0.9 | 61.1±1.0 | 79.6±1.2 | 62.5±8.1 | 48.9±2.8 | 13.3±1.1 |
| CDANN (Li et al., 2018c) | 73.0±1.3 | 67.8±2.0 | 75.0±0.6 | 65.2±1.1 | 70.3±0.5 | 63.9±1.0 | 83.5±0.8 | 50.0±4.2 | 43.9±4.7 | 20.4±3.1 |
| MTL (Blanchard et al., 2021) | 79.4±0.8 | 70.8±0.6 | 81.9±0.8 | 67.8±1.4 | 75.0±0.7 | 67.7±1.4 | 85.2±0.7 | 73.8±1.6 | 61.1±2.8 | 12.4±4.0 |
| SagNet (Nam et al., 2021) | 79.4±1.8 | 71.2±0.7 | 83.4±2.4 | 66.5±2.1 | 75.1±1.6 | 66.5±2.1 | 85.5±0.9 | 77.1±5.0 | 57.8±4.3 | 13.0±3.4 |
| Fish (Shi et al., 2021) | 80.1±1.9 | 70.2±0.2 | 84.4±0.9 | 66.3±0.5 | 75.3±0.5 | 66.3±0.5 | 85.8±0.2 | 73.3±3.9 | 61.1±3.0 | 13.7±3.3 |
| Focal (Lin et al., 2017) | 80.9±0.7 | 71.6±1.6 | 84.4±1.3 | 66.1±1.7 | 75.7±0.4 | 65.3±1.1 | 85.7±0.3 | 76.2±3.9 | 68.9±3.2 | 12.6±1.9 |
| CBLoss (Cui et al., 2019) | 84.9±0.6 | 78.0±1.2 | 80.7±0.3 | 68.3±2.0 | 78.0±0.4 | 68.3±2.0 | 85.0±0.1 | 89.2±1.2 | 83.9±2.5 | 9.3±3.9 |
| LDAM (Cao et al., 2019) | 83.0±0.9 | 70.6±0.6 | 81.3±1.1 | 64.1±1.4 | 74.7±0.9 | 64.1±1.4 | 85.1±0.6 | 70.8±3.5 | 67.8±1.2 | 11.1±2.4 |
| BSoftmax (Ren et al., 2020) | 83.5±2.1 | 75.5±0.4 | 82.1±0.7 | 65.6±1.3 | 76.7±1.0 | 65.6±1.3 | 83.4±0.8 | 90.8±0.9 | 78.3±3.9 | 12.6±2.4 |
| SSP (Yang & Xu, 2020) | 82.6±1.3 | 80.7±1.8 | 83.2±0.6 | 67.3±0.4 | 78.5±0.7 | 67.3±0.4 | 85.5±1.0 | 87.8±0.9 | 82.6±1.2 | 13.2±2.8 |
| CRT (Kang et al., 2020) | 89.0±0.1 | 81.8±0.3 | 85.8±0.3 | 70.0±0.4 | 81.6±0.1 | 70.0±0.4 | 89.7±0.2 | 90.4±0.3 | 83.9±0.5 | 12.9±0.0 |
| BoDAr (Yang et al., 2022) | 86.7±0.7 | 74.1±1.1 | 85.2±0.7 | 67.3±0.3 | 78.6±0.4 | 68.5±0.3 | 86.4±0.1 | 85.0±1.0 | 80.0±0.9 | 13.7±2.1 |
| BoDA-Mr (Yang et al., 2022) | 87.8±0.9 | 76.5±0.9 | 82.2±0.3 | 71.3±0.4 | 79.4±0.6 | 71.3±0.4 | 88.4±0.3 | 76.2±2.7 | 88.3±1.6 | 14.4±1.4 |
| BoDAr,c (Yang et al., 2022) | 88.3±0.6 | 82.9±0.5 | **89.3**±0.9 | 68.5±0.6 | 82.3±0.3 | 68.5±0.6 | 89.2±0.2 | **92.5**±0.9 | 88.3±1.2 | 21.3±0.7 |
| BoDA-Mr,c (Yang et al., 2022) | **90.4**±0.3 | 81.2±0.7 | 85.8±0.4 | **74.6**±0.7 | 83.0±0.4 | **74.6**±0.7 | 89.2±0.2 | 91.2±0.6 | **91.7**±2.0 | **21.7**±1.4 |
| **Ours** | 89.9±0.5 | **84.7**±0.6 | 87.6±0.8 | 73.5±0.6 | **83.9**±0.2 | 73.5±0.6 | **91.4**±0.3 | **92.5**±0.6 | 88.3±3.4 | 21.4±1.2 |

Table 12: Accuracy (%) under the multi-domain long-tailed (MDLT) setting on **DomainNet-MLT**. Results are sourced from their original papers, and are reported by domain and by shot.

| Algorithm | Accuracy (by domain) | | | | | | | | Accuracy (by shot) | | | |
|---|---|---|---|---|---|---|---|---|---|---|---|---|
| | Clip | Info | Paint | Quick | Real | Sketch | Average | Worst | Many | Medium | Few | Zero |
| ERM (Vapnik, 1998) | 68.6±0.1 | 29.4±0.3 | 57.1±0.2 | 62.8±0.3 | 72.1±0.2 | 61.7±0.2 | 58.6±0.2 | 29.4±0.3 | 66.0±0.1 | 56.1±0.1 | 35.9±0.5 | 27.6±0.3 |
| IRM (Arjovsky et al., 2019) | 66.7±0.2 | 27.6±0.1 | 56.0±0.2 | 60.1±0.1 | 72.0±0.0 | 60.2±0.2 | 57.1±0.1 | 27.6±0.1 | 64.7±0.1 | 54.3±0.3 | 33.5±0.3 | 25.8±0.3 |
| GroupDRO (Sagawa* et al., 2020) | 60.1±0.2 | 25.9±0.2 | 50.3±0.1 | 63.9±0.2 | 64.9±0.2 | 56.7±0.3 | 53.6±0.1 | 25.9±0.2 | 61.8±0.1 | 49.1±0.3 | 30.7±0.7 | 22.0±0.1 |
| Mixup (Zhang et al., 2018) | 67.6±0.2 | 28.7±0.0 | 56.4±0.2 | 60.0±0.4 | 72.1±0.1 | 60.9±0.1 | 57.6±0.1 | 28.7±0.0 | 64.9±0.2 | 54.5±0.1 | 35.6±0.2 | 27.3±0.3 |
| MLDG (Li et al., 2018a) | 68.0±0.2 | 28.7±0.1 | 57.2±0.1 | 61.6±0.2 | 73.3±0.1 | 61.9±0.2 | 58.5±0.0 | 28.7±0.1 | 66.0±0.1 | 55.7±0.1 | 35.3±0.2 | 26.9±0.3 |
| CORAL (Sun & Saenko, 2016) | 69.1±0.3 | 30.1±0.4 | 57.8±0.2 | 63.4±0.2 | 72.8±0.2 | 63.3±0.3 | 59.4±0.1 | 30.1±0.4 | 66.4±0.1 | 57.1±0.0 | 37.7±0.6 | 29.9±0.2 |
| MMD (Li et al., 2018b) | 66.1±0.1 | 27.2±0.2 | 55.9±0.1 | 59.3±0.2 | 71.9±0.1 | 60.0±0.2 | 56.7±0.0 | 27.2±0.2 | 64.2±0.1 | 54.0±0.0 | 33.9±0.2 | 25.4±0.2 |
| DANN (Ganin et al., 2016) | 65.5±0.3 | 26.9±0.4 | 55.2±0.1 | 57.4±0.2 | 70.6±0.1 | 59.0±0.2 | 55.8±0.1 | 26.9±0.4 | 63.0±0.1 | 52.7±0.1 | 34.2±0.4 | 26.8±0.4 |
| CDANN (Li et al., 2018c) | 65.9±0.1 | 27.7±0.1 | 55.3±0.1 | 57.6±0.2 | 70.9±0.2 | 58.7±0.1 | 56.0±0.1 | 27.7±0.1 | 63.2±0.0 | 52.7±0.2 | 34.3±0.5 | 27.6±0.1 |
| MTL (Blanchard et al., 2021) | 68.2±0.2 | 29.3±0.2 | 57.3±0.1 | 62.1±0.1 | 72.9±0.1 | 61.8±0.2 | 58.6±0.1 | 29.3±0.2 | 65.9±0.1 | 56.0±0.4 | 35.4±0.1 | 28.2±0.3 |
| SagNet (Nam et al., 2021) | 68.5±0.1 | 29.4±0.2 | 57.8±0.2 | 62.1±0.2 | 73.3±0.1 | 62.4±0.1 | 58.9±0.0 | 29.4±0.2 | 66.3±0.1 | 56.4±0.0 | 36.2±0.3 | 27.2±0.4 |
| Fish (Shi et al., 2021) | 68.7±0.1 | 29.1±0.1 | 58.4±0.1 | 64.1±0.1 | 73.9±0.1 | 63.7±0.1 | 59.6±0.1 | 29.1±0.1 | 67.1±0.1 | 57.2±0.1 | 36.8±0.4 | 27.8±0.3 |
| Focal (Lin et al., 2017) | 67.6±0.2 | 27.5±0.1 | 56.5±0.3 | 62.3±0.3 | 71.7±0.3 | 61.4±0.3 | 57.8±0.2 | 27.5±0.1 | 65.2±0.2 | 55.1±0.2 | 35.8±0.1 | 26.3±0.1 |
| CBLoss (Cui et al., 2019) | 68.3±0.2 | 30.1±0.1 | 57.8±0.1 | 60.8±0.1 | 73.3±0.2 | 63.3±0.1 | 58.9±0.1 | 30.1±0.1 | 64.3±0.1 | 61.0±0.3 | 42.5±0.4 | 28.1±0.2 |
| LDAM (Cao et al., 2019) | 68.8±0.2 | 29.2±0.2 | 57.1±0.1 | 65.0±0.2 | 73.2±0.1 | 63.1±0.1 | 59.3±0.2 | 29.2±0.2 | 66.6±0.0 | 57.0±0.0 | 37.1±0.2 | 27.8±0.3 |
| BSoftmax (Ren et al., 2020) | 68.5±0.1 | 29.9±0.1 | 57.8±0.1 | 60.5±0.3 | 73.4±0.1 | 63.3±0.0 | 58.9±0.1 | 29.9±0.1 | 64.3±0.1 | 60.9±0.3 | 42.4±0.6 | 28.2±0.1 |
| SSP (Yang & Xu, 2020) | 69.7±0.1 | 31.6±0.2 | 58.8±0.1 | 59.7±0.3 | 73.9±0.1 | 64.2±0.1 | 59.7±0.0 | 31.6±0.2 | 64.3±0.1 | 62.6±0.1 | 45.0±0.3 | 30.5±0.0 |
| CRT (Kang et al., 2020) | 70.0±0.2 | 31.6±0.1 | 59.2±0.2 | 64.0±0.1 | 73.4±0.1 | 64.4±0.1 | 60.4±0.2 | 31.6±0.1 | 66.8±0.0 | 61.6±0.1 | 45.7±0.1 | 29.7±0.1 |
| BoDAr (Yang et al., 2022) | 70.0±0.1 | 32.6±0.1 | 59.1±0.2 | 61.2±0.4 | 73.3±0.1 | 64.1±0.1 | 60.1±0.2 | 32.6±0.1 | 65.7±0.2 | 60.6±0.1 | 42.6±0.3 | 30.5±0.2 |
| BoDA-Mr (Yang et al., 2022) | 70.6±0.1 | 32.2±0.2 | 57.7±0.3 | 65.5±0.9 | 73.2±0.1 | 64.5±0.1 | 60.1±0.2 | 32.2±0.2 | 65.9±0.2 | 60.7±0.1 | 42.9±0.3 | 30.0±0.1 |
| BoDAr,c (Yang et al., 2022) | 72.0±0.2 | 33.4±0.1 | 60.7±0.2 | 63.6±0.2 | 74.6±0.1 | 65.5±0.2 | 61.7±0.1 | 33.4±0.1 | 67.0±0.1 | 62.7±0.1 | 46.0±0.2 | **32.2**±0.3 |
| BoDA-Mr,c (Yang et al., 2022) | 71.8±0.1 | 33.3±0.1 | 60.8±0.1 | 63.7±0.3 | 74.6±0.1 | 65.8±0.2 | 61.7±0.2 | 33.3±0.1 | 67.0±0.1 | **63.0**±0.3 | **46.6**±0.4 | 31.8±0.2 |
| **Ours** | **72.8**±0.0 | **33.8**±0.1 | **63.4**±0.2 | **67.0**±0.2 | **78.9**±0.1 | **67.2**±0.0 | **63.8**±0.1 | **33.8**±0.1 | **71.0**±0.1 | 62.3±0.1 | 41.9±0.2 | 31.6±0.3 |

Table 13: Accuracy (%) under the multi-domain long-tailed (MDLT) setting on **DomainNet126-MLT**. Results are sourced from their original papers, and are reported by domain and by shot.

| Algorithm | Accuracy (by domain) | | | | | | Accuracy (by shot) | | | |
|---|---|---|---|---|---|---|---|---|---|---|
| | C | P | R | S | Average | Worst | Many | Medium | Few | Zero |
| ERM (Vapnik, 1998) | 78.9±0.4 | 71.1±0.3 | 85.6±0.1 | 78.6±0.3 | 78.6±0.1 | 71.1±0.3 | 84.1±0.1 | 78.5±0.2 | 62.7±0.6 | 48.3±0.5 |
| IRM (Arjovsky et al., 2019) | 78.4±0.0 | 72.9±0.0 | 86.5±0.0 | 78.7±0.0 | 79.1±0.0 | 72.9±0.0 | 84.6±0.0 | 79.4±0.0 | 63.0±0.0 | 47.0±0.0 |
| GroupDRO (Sagawa* et al., 2020) | 77.4±0.3 | 70.0±0.5 | 86.6±0.1 | 77.8±0.2 | 77.9±0.3 | 70.0±0.5 | 83.9±0.2 | 77.5±0.2 | 60.5±1.7 | 50.0±1.1 |
| Mixup (Zhang et al., 2018) | 80.6±0.2 | 73.1±0.3 | 86.9±0.1 | 80.2±0.2 | 80.2±0.1 | 73.1±0.3 | 85.3±0.0 | 80.6±0.1 | 65.6±1.2 | 49.5±0.1 |
| MLDG (Li et al., 2018a) | 79.0±0.1 | 70.9±0.2 | 85.9±0.1 | 78.2±0.1 | 78.5±0.0 | 70.9±0.2 | 84.1±0.1 | 78.7±0.2 | 62.4±0.3 | 47.8±0.6 |
| CORAL (Sun & Saenko, 2016) | 79.4±0.2 | 73.0±0.1 | 86.6±0.2 | 79.3±0.1 | 79.6±0.1 | 73.0±0.1 | 84.9±0.1 | 79.6±0.1 | 64.0±0.2 | 51.7±0.9 |
| MMD (Li et al., 2018b) | 79.2±0.0 | 72.3±0.0 | 86.4±0.0 | 79.5±0.0 | 79.3±0.0 | 72.3±0.0 | 84.5±0.0 | 79.8±0.0 | 63.9±0.0 | 49.5±0.0 |
| DANN (Ganin et al., 2016) | 74.7±0.5 | 68.8±0.2 | 82.1±0.3 | 75.2±0.1 | 75.2±0.2 | 68.8±0.2 | 80.7±0.3 | 74.5±0.3 | 60.5±0.8 | 47.7±0.9 |
| CDANN (Li et al., 2018c) | 75.8±0.5 | 69.1±0.3 | 82.0±0.3 | 75.4±0.3 | 75.6±0.2 | 69.1±0.3 | 80.6±0.2 | 75.5±0.2 | 61.3±0.7 | 48.0±0.7 |
| MTL (Blanchard et al., 2021) | 78.1±0.1 | 70.6±0.4 | 85.9±0.2 | 77.8±0.2 | 78.1±0.2 | 70.6±0.4 | 83.9±0.1 | 78.2±0.4 | 61.4±0.7 | 45.5±0.6 |
| SagNet (Nam et al., 2021) | 79.0±0.1 | 71.5±0.4 | 85.5±0.3 | 79.0±0.2 | 78.8±0.2 | 71.5±0.4 | 84.1±0.2 | 79.2±0.3 | 61.3±0.8 | 49.4±1.0 |
| Fish (Shi et al., 2021) | 79.0±0.4 | 70.7±0.3 | 86.2±0.1 | 78.5±0.2 | 78.6±0.1 | 70.7±0.3 | 84.2±0.2 | 78.6±0.4 | 63.0±0.3 | 47.6±1.0 |
| Focal (Lin et al., 2017) | 78.1±0.3 | 70.0±0.2 | 85.1±0.3 | 78.1±0.2 | 77.8±0.2 | 70.0±0.2 | 83.6±0.1 | 77.8±0.2 | 61.8±0.5 | 45.8±0.4 |
| CBLoss (Cui et al., 2019) | 79.0±0.2 | 71.3±0.4 | 85.8±0.3 | 79.0±0.1 | 78.8±0.2 | 71.3±0.4 | 83.3±0.3 | 80.0±0.3 | 65.2±0.1 | 48.8±0.8 |
| LDAM (Cao et al., 2019) | 79.0±0.1 | 71.1±0.1 | 86.2±0.2 | 78.2±0.3 | 78.6±0.1 | 71.1±0.1 | 83.8±0.1 | 79.2±0.1 | 63.2±1.1 | 49.2±1.1 |
| BSoftmax (Ren et al., 2020) | 79.9±0.3 | 71.9±0.4 | 86.0±0.1 | 79.9±0.0 | 79.4±0.2 | 71.9±0.4 | 83.2±0.0 | 80.4±0.1 | **68.1**±0.9 | **55.0**±0.8 |
| CRT (Kang et al., 2020) | 80.2±0.2 | 72.5±0.1 | 87.2±0.1 | 80.8±0.1 | 80.2±0.0 | 72.5±0.1 | 85.3±0.0 | 80.7±0.1 | 65.1±0.4 | 51.0±0.5 |
| BoDA (Yang et al., 2022) | 80.0±0.2 | 73.1±0.1 | 86.4±0.2 | 79.4±0.2 | 79.7±0.0 | 73.1±0.1 | 84.5±0.1 | 80.4±0.1 | 65.8±0.7 | 51.7±1.2 |
| **Ours** | **81.0**±0.2 | **74.2**±0.2 | **88.4**±0.2 | **80.9**±0.1 | **81.1**±0.2 | **74.2**±0.2 | **86.5**±0.1 | **81.6**±0.2 | 64.6±0.4 | 51.0±0.4 |

# I Computational Cost Analysis

We analyze the computational overhead introduced by the RC-Align framework compared to ERM and MLDG. We measured the runtime and peak memory usage on the OfficeHome dataset (5000 steps) in a standardized environment (NVIDIA RTX 3090). The results are summarized in Table 14.

Table 14: Computational cost comparison on OfficeHome. Avg. DG Acc. refers to the results reported in Table 1.

| Method | Runtime (H:MM:SS) | Relative Runtime | Max Memory (GB) | Avg. DG Acc. |
|---|---|---|---|---|
| ERM | 1:16:46 | 1.00x | 1.91 | 68.9 |
| MLDG | 1:28:51 | 1.16x | 1.88 | 69.2 |
| RC-Align | 1:51:03 | 1.45x | 2.31 | 73.0 |

The experimental results indicate that RC-Align requires approximately $1.45\times$ the training time of ERM. This overhead is comparatively lower than standard bilevel optimization methods, due to the adoption of the FO-MAML approximation. Considering the performance gain of $+4.1\%p$ over ERM, we consider this increased training time to represent a reasonable trade-off.

# J    LIMITATIONS AND FUTURE RESEARCH

While this work introduces a novel theoretical framework and the RC-Align algorithm for domain generalization under compound distribution shifts, several limitations exist that suggest directions for future research.

**Architectural Dependence and Scalability.**    Our experimental evaluation was conducted using a ResNet-50 backbone pre-trained on ImageNet, following standard protocols in the domain generalization literature. While this ensures fair comparison with existing work, it remains unverified whether the benefits of RC-Align are specific to CNN architectures or if they extend to more recent architectures like Vision Transformers (ViTs). Furthermore, its effectiveness in settings where models are trained from scratch has not been explored. Future work should validate the generalizability of the proposed methodology across a diverse range of backbone architectures.

**Computational Overhead of Meta-Learning.**    RC-Align employs a MAML-style meta-learning framework involving a nested optimization process. Although the use of the First-Order MAML (FO-MAML) approximation mitigates the cost to a reasonable trade-off (approximately $1.45\times$ that of ERM), the method inevitably demands higher computational resources than standard training. This additional computational burden remains a constraint that should be considered, particularly when scaling to massive datasets or operating in resource-limited environments.

**Centroid Estimation in Long-Tail Scenarios.**    The proposed Domain-Class Distribution Alignment (DA) loss relies on the accurate estimation of class-wise centroids from the source domains. In this study, centroids are estimated on-the-fly using mini-batches for computational efficiency within the meta-learning loop. However, this approach can introduce noise and instability in the estimates, particularly for tail classes in the Multi-Domain Long-Tail (MDLT) setting where data is sparse. Inaccurate centroid estimation could undermine the effectiveness of feature alignment, necessitating the exploration of more robust centroid estimation techniques that do not significantly increase computational complexity.

**Assumptions in Theoretical Analysis.**    The theoretical analysis of the meta-learning procedure's optimization dynamics (Section 3.3) relies on strong assumptions such as L-smoothness and the Polyak–Lojasiewicz (PL) condition to characterize convergence behavior. As acknowledged in the paper, the PL condition may not hold globally for the highly non-convex loss landscapes of deep neural networks. Therefore, the theoretical guarantees regarding optimization should be interpreted as holding under local or idealized settings.

**Theoretical Scope on Domain Interpolation.**    Consistent with prevalent assumptions in Domain Generalization literature Muandet et al. (2013); Blanchard et al. (2021); Wang et al. (2022); Rosenfeld et al. (2022), our theoretical framework primarily addresses scenarios where target domains lie within the support of source domains (i.e., interpolation). This focus reflects a standard characteristic of mixture-based DG analysis. Addressing generalization to extrapolated targets often necessitates modeling causal mechanisms or mitigating negative transfer effects Arjovsky et al. (2019); Mahajan et al. (2021); Sheth et al. (2022); Lv et al. (2022), which presents a challenge orthogonal to our current problem setting. We identify robustness against extrapolation as a promising direction for future research.

## K  DISCLOSURE OF LARGE LANGUAGE MODEL (LLM) USAGE

The development of this manuscript was assisted by a Large Language Model (LLM). The LLM's role was confined to that of a general-purpose tool, and it did not contribute to the core research ideation or experimental design. The authors retained full intellectual control and responsibility for the final content.

Specifically, the LLM was utilized in the following capacities:

- **Manuscript Writing and Grammar Correction:** The LLM was employed to improve the clarity, flow, and grammatical accuracy of the manuscript. This included refining sentence structures and correcting typographical errors.

- **Review of Mathematical Derivations:** During the process of proving theoretical results, the LLM was used as a tool to review and verify the step-by-step expansion of mathematical expressions, ensuring the derivations were sound and free of algebraic errors.

- **Code Generation and Analysis:** The LLM provided assistance in generating boilerplate code for experimental setups and developing scripts for the analysis of experimental results. All generated code was thoroughly reviewed, tested, and modified by the authors to ensure its correctness and suitability for the research.

The authors take full responsibility for all content presented in this paper, including any text, mathematical proofs, or code influenced by the LLM.

