# OpenReview forum: "Robust Domain Generalization under Divergent Marginal and Conditional Distributions"
_ICLR.cc/2026/Conference — Submitted to ICLR 2026_

### Official Review · Reviewer_Fem2 · 2025-10-21

**Soundness:** 3
**Presentation:** 3
**Contribution:** 2
**Rating:** 4
**Confidence:** 4

**Summary:**

The paper studies domain generalization under a compound shift in both $P(Y)$ and $P(Z|Y)$. A risk bound is provided, which decomposes the generalization gap into 2 parts: a prior shift term consisting of class distribution weighted risks, and a feature shift term consisting of Wasserstein distances between class-conditional feature distributions. Building on this, the paper proposes RC-Align, a meta-learning method that regularizes on a domain-class distribution alignment loss. Experiments on standard DG and MDLT benchmarks show strong average and worst-domain performance.

**Strengths:**

* This paper conducts a solid theoretical analysis of the domain generalization error. The decomposition into prior and feature shift terms is intuitive and insightful. The algorithm design is also closely connected with these theoretical impressions.
* Experimental results show decent performance improvement on both standard DG and MDLT benchmarks.

**Weaknesses:**

* From my current understanding, the current theoretical framework cannot guarantee generalization on an arbitrary target domain. In Theorem 3, the performance depends on the Wasserstein distance between the target data distribution and its best approximation via interpolation between source domains. Hence, it only guarantees generalization under the condition that the target domain is an interpolation of source domains, which is known to be well-resolved by ERM. The cases where the target domain is an extrapolation cannot be handled by the current results. Also, the idea of building generalization bounds via controlling the inter-domain feature distribution alignment is not new.
* The current hyperparameter selection scheme does not follow the DomainBed standard, which requires a sweep among a predefined joint distribution of all hyperparameters. The learning rate for RC-Align is fixed to 5e-5 as stated, which differs from the standard settings and may result in unfair comparison.
* The meta-learning scheme and the DA loss design are both borrowed from existing works. The methodological contribution is thus somewhat limited to a combination of existing methods. Also, the meta-learning scheme may introduce a significant computational burden compared to other baselines. This weakness is already acknowledged by the authors, which is good. However, there is no running time/memory comparison provided in the current version, so it is unclear whether the performance improvement is worthwhile with increased computational costs.

**Questions:**

* According to Theorem 3, it seems that it suffices to minimize the empirical risk $R_{D_i}$ on each source domain to achieve generalization, since the second term is intractable under the DG setting. I don't see a clear motivation for why we need additional upper bounds for $R_{D_i}$ in Theorems 1-2, since it can be directly optimized via ERM. Can the authors further explain this point?

---

> ### Author Response · Authors · 2025-11-19
>
> ### Response to Reviewer Fem2
>
> *   **W1. The current theoretical framework cannot handle extrapolation. It only guarantees generalization under interpolation, which is known to be well-resolved by ERM. The idea of alignment bounds is not new.**
>
>      A1. Please refer to '3. Theoretical Limitations and Guarantees' in the Common Responses. We clarify that (R1) ERM does *not* resolve generalization under compound shifts even with interpolation; (R2) our alignment bound is novel as it provides a risk-aware decomposition under compound shifts; and (R3) our methodology provides higher robustness in extrapolation scenarios than ERM.
>
> *   **W2. The hyperparameter selection scheme does not follow the DomainBed standard. The learning rate for RC-Align is fixed to 5e-5.**
>
>     A2. We strictly adhered to one of DomainBed's standard model selection protocols, **"Training-domain validation"** (Gulrajani & Lopez-Paz, 2021). The learning rate mentioned (5e-5) is not a fixed value, but the **optimal value selected** through this validation procedure within the standard search space (Appendix E.2). All baselines were tuned with the same protocol, ensuring a fair comparison.
>
> *   **W3. The methodological contribution is limited, and there is no running time/memory comparison.**
>
>      A3. (For methodological contribution, please refer to '1. Regarding Novelty'.) The concern regarding computational cost is valid. As requested, we measured the runtime and peak memory usage on the OfficeHome dataset (5000 steps) in the same environment (NVIDIA RTX 3090).
>
>     | Method | Runtime (H:MM:SS) | Relative Runtime | Max Memory (GB) | Avg. DG Acc. (Table 1) |
>     | :---: | :---: | :---: | :---: | :---: |
>     | ERM | 1:16:46 | 1.00x | 1.91 | 68.9 |
>     | MLDG | 1:28:51 | 1.16x | 1.88 | 69.2 |
>     | **RC-Align** | **1:51:03** | **1.45x** | **2.31** | **73.0** |
>
>     The results show that RC-Align takes approximately 1.45x the runtime of ERM. This overhead is significantly lower than typical bilevel optimization, thanks to the FO-MAML approximation and efficient implementation. Given the substantial performance improvement achieved by RC-Align (+4.1%p over ERM), we believe this moderate increase in training time is well justified.
>
> *   **Q1. It seems sufficient to minimize the empirical risk (ERM) according to Theorem 3. Why are additional upper bounds in Theorems 1-2 needed?**
>
>     A1.Theorem 3 bounds the target risk by (Source Mixture Risk + Distribution Distance).
>     ERM only minimizes the source mixture risk term and leaves the distribution-shift–induced excess risk (the prior-mismatch and feature-mismatch terms in Theorems 1–2) completely uncontrolled.
>
>     Theorems 1 and 2 are essential because they characterize these shift-sensitive terms and show how they can be effectively upper-bounded by optimizable quantities (class-wise risks and the DA loss).
>     Therefore, they provide the practically optimizable objectives needed for  domain generalization beyond ERM.

---

> > ### Comment · Reviewer_Fem2 · 2025-11-26
> >
> > Thanks for the response. I'm still confused about the implications given by Theorems 1-2. They upper bound the training-domain risk $R_{D_i}$, which can be directly minimized by ERM. I don't see why we need to optimize an upper bound of this risk rather than directly optimize the risk itself. For DG, the ultimate goal is to minimize $R_T$, which in your theory is decomposed into a weighted sum of $R_{D_i}$ and another target-domain shift term. The first one can be directly optimized by ERM, while the second one is an intrinsic property of the data distribution and cannot be optimized by any means. To me, minimizing the prior and feature shift terms can be a sufficient condition to minimize $R_{D_i}$, but completely unnecessary here.

---

> > > ### Author Response · Authors · 2025-11-27
> > >
> > > Dear Reviewer Fem2,
> > >
> > > Thank you for the insightful follow-up questions. We appreciate the opportunity to clarify these aspects.
> > >
> > > In our setting, the ultimate goal is to minimize the risk on an unseen target domain, whose data are not available during training and thus cannot be directly handled by ERM.
> > > We therefore derive an upper bound on this target risk in terms of source-domain quantities and representation-dependent shift terms, and minimize this bound as a tractable surrogate that indirectly controls the target-domain risk.
> > >
> > > To clarify further, we have added detailed explanations.
> > >
> > > ---
> > >
> > > ### 1. Clarification of Notation: Target (Unseen) Domain Risk
> > >
> > > It seems there is a misunderstanding regarding the definition of $R_{D_i}$ in the context of Theorems 1-3. The reviewer states that $R_{D_i}$ is the "training-domain risk, which can be directly minimized by ERM." This interpretation may not fully align with our analytical framework:
> > >
> > > * $D_i$ is designated as the **query domain** (i.e., the held-out, unseen target domain).
> > > * $D_{-i}$ represents the **support mixture** (i.e., the source/training domains).
> > >
> > > This can be found and is detailed in Section 3 (Page 3) of our paper.
> > >
> > > ### 2. Controlling the Shift-Induced Excess Risk: From Intractable Theory to Optimizable Objectives via Theorems 1–2.
> > >
> > > We would like to clarify that the shift-dependent components of the bound are indeed optimizable. Below, we explain how these shift-sensitive terms can be controlled through representation learning, and why Theorems 1–2 are essential for making this optimization feasible in practice.
> > >
> > > **A. The Shift-Induced Excess Risk is Controllable in the Feature Space**
> > >
> > > Although the shift in the input distribution $P(X, Y)$ is intrinsic to the environment, the resulting discrepancy in the **feature-space distributions** $P(Z, Y)$ (where $Z = f_\\theta(X)$) is governed by the learned representation $f_\\theta$. By aligning class-conditional feature distributions across domains, the model can actively reduce the **shift-induced excess risk** appearing in the upper bound. In this sense, the representation plays a central role in controlling the terms that ERM alone cannot address.
> > >
> > > **B. Why Theorems 1–2 Are Necessary: From Intractable W1 to a Tractable Alignment Objective**
> > >
> > > To control these shift-sensitive terms, we must make them quantifiable and optimizable. Theorem 1 decomposes the target-domain risk into:
> > >
> > > $$
> > > R_{D_i}(\\Theta) \\le R_{D_{-i}}(\\Theta) + \\sum_{c \\in C} | \\pi_i(c)-\\pi_{-i}(c) | \\cdot R_{-i,c}(\\Theta) + L_\\ell \\sum_{c \\in C} \\pi_i(c)\\, W_1(P_{i,c}, P_{-i,c})
> > > $$
> > >
> > > (Terms correspond to: Target Risk $\\le$ Source Risk + Prior Shift + Feature Shift)
> > >
> > > However, directly minimizing the **feature shift term**, which involves the 1-Wasserstein distance $W_1(P_{i,c}, P_{-i,c})$, is fundamentally intractable due to:
> > >
> > > 1. **Inaccessibility of Target Data:** ($P_{i,c}$ is unobserved), and
> > > 2. **Computational Complexity:** in high-dimensional spaces.
> > >
> > > **Theorem 2** addresses this challenge by proving that the intractable Wasserstein ($W_1$) term can be upper-bounded by a **tractable, differentiable surrogate** — the Domain-Class Distribution Alignment loss $L_{DA}$.
> > > Unlike $W_1$, $L_{DA}$ is:
> > >
> > > * **Efficient:** It relies on centroid-based distances rather than solving optimal transport problems.
> > > * **Differentiable:** It is structured as a smooth contrastive loss, allowing seamless integration into gradient-based optimization (SGD).
> > > * **Computable using source domains alone:** It does not require target-domain samples.
> > >
> > > Thus, Theorems 1–2 collectively transform an intractable theoretical bound into a **practical and optimizable objective** that can be minimized during training.

---

> ### Comment · Reviewer_Fem2 · 2025-11-27
>
> Thank you for the response. It appears that the definition of source and target domains is somewhat ambiguous. Here is my interpretation of the procedure for applying your meta-learning algorithm:
>
> * The dataset consists of $N$ distinct domains, and a Leave-One-Domain-Out (LODO) evaluation protocol is employed.
>   * For each held-out domain, $K = N - 1$ domains are used for training the model.
>   * Within these $K$ domains, another round of leave-one-out is performed: for each domain treated as a target, the model is trained to transfer from the remaining $K - 1$ source domains using Algorithm 1.
>   * The resulting model is then evaluated on the originally held-out domain, transferring from all $K$ domains.
> * The final performance is computed as the average over all $N$ held-out domains.
>
> In this setup, each domain $D_1, \ldots, D_K$ is involved in training, and thus $R_{D_i}$ can be explicitly minimized by the training algorithm. Please correct me if this does not accurately reflect your implementation.

---

> > ### Author Response · Authors · 2025-11-29
> >
> > Dear Reviewer Fem2,
> >
> > Thank you for the insightful questions regarding the interpretation of our procedure and the necessity of our theoretical bounds.
> >
> > ### Summary of Key Points
> >
> > Theorem 3 decomposes the target risk $R_{T}(\\Theta)$ into the weighted source risks ($\\sum \\pi_i R_{D_i}(\\Theta)$) and the distribution shift term ($L_l W_1(T, T_\\pi)$). Since the target risk is upper-bounded by the sum of these two terms, minimizing only the source risk (as ERM does) is insufficient.
> >
> > In deep networks, ERM often lowers the source risk by learning source-specific shortcuts, which simultaneously enlarges the shift term (which depends on the learned representation $\\Theta$). The two terms are therefore **coupled**, and controlling only one does not guarantee low target risk.
> >
> > Because the true target distribution $T$ is unknown, the shift term in Theorem 3 is **intractable**. Our method provides a **tractable surrogate** (derived in Theorems 1 & 2) by letting each available domain ($D_i$) act as a proxy target in a meta-learning setup. This enables us to jointly minimize source risk and an estimable shift term ($L_{DA}$). This directly operationalizes the bound rather than assuming ERM will keep the shift small.
> >
> > In summary, the shift term is not redundant: it captures the part of the target risk that ERM cannot control, and explicitly minimizing it leads to the improved generalization observed in our experiments.

---

> > > ### Author Response · Authors · 2025-11-29
> > >
> > > ### Detailed Clarification
> > >
> > > To clarify more in detail, we have provided extra clarification here.
> > >
> > > ### 1. Clarification: The Role of $D_i$ in the Meta-Learning Framework
> > >
> > > While $D_i$ is available during the overall training process, our framework explicitly treats it as a **virtual query (target) domain** within a specific episode to simulate the unseen nature of the true target $T$.
> > >
> > > * **ERM Approach:** Treats $D_i$ simply as another training source and minimizes the risk on $D_i$ directly as part of the aggregate source risk.
> > > * **Our Approach:** Treats $D_i$ as a held-out proxy for $T$. We use the simulated split ($D_{-i}$ vs $D_i$) to measure and minimize the distribution shift. This allows us to learn representations that generalize to truly unseen domains $T$.
> > >
> > > In the context of the analysis in Theorems 1 & 2, $R_{D_i}$ specifically refers to the risk on this **virtual target domain**, which is held out during the adaptation step.
> > >
> > > ### 2. The Limitations of ERM regarding Theorem 3
> > >
> > > The reviewer suggests that based on Theorem 3, minimizing $R_{D_i}$ via ERM should be sufficient, and that the shift term is an "intrinsic property" that cannot be optimized. This overlooks the dependency of the shift term on the model parameters $\\Theta$.
> > >
> > > Theorem 3 states:
> > >
> > > $$
> > > R_T(\\Theta) \\le \\sum \\pi_i R_{D_i}(\\Theta) + L_l W_1(T, T_\\pi)
> > > $$
> > >
> > > (i.e., Target Risk $\\le$ Source Risk + Shift Term)
> > >
> > > The critical issue is that these two terms are **coupled via the representation $\\Theta$**. The Shift Term is measured in the feature space and is therefore optimizable.
> > >
> > > * ERM focuses solely on minimizing the **Source Risk**.
> > > * However, ERM often achieves this by learning features highly specific to the source domains (e.g., spurious correlations), which paradoxically **increases the Shift Term**.
> > >
> > > Thus, ERM performs an incomplete optimization of the bound in Theorem 3, often leading to suboptimal generalization. Furthermore, the Shift Term in Theorem 3 depends on the unknown target $T$, making the full bound **intractable** to optimize directly.
> > >
> > > ### 3. Why We Optimize the Bound from Theorems 1 & 2
> > >
> > > Our strategy (Theorems 1 & 2 $\\rightarrow$ C.6) addresses this by replacing the intractable Shift Term in Theorem 3 with a **tractable surrogate** derived from the meta-training splits ($D_i$ vs $D_{-i}$).
> > >
> > > * Theorems 1 and 2 analyze the generalization gap between the virtual sources and the virtual target, leading to the Domain Alignment loss ($L_{DA}$) as a tractable proxy for the feature shift.
> > > * The resulting combined bound (C.6) is **fully optimizable using only source data**. By minimizing this surrogate bound, we jointly control **both** the risk and the shift.
> > >
> > > Unlike ERM, which ignores the shift, our method actively penalizes feature distributions that diverge between domains (via $L_{DA}$). This ensures that minimizing the source risk does not come at the cost of increased distribution shift.
> > >
> > > **Conclusion:** We optimize the bound derived from Theorems 1 and 2 because it allows us to minimize the total upper bound (Risk + Shift) simultaneously using tractable objectives. In contrast, ERM optimizes only the risk component and ignores the crucial, representation-dependent shift component, leading to a higher realized target risk $R_T$.
> > >
> > > ### 4. Why Meta-Learning Differs from ERM: A Domain-Invariant, Task-Level Perspective
> > >
> > > Although all domains are available during training, meta-learning optimizes a *different* objective from ERM.
> > > ERM minimizes a single aggregated risk over all sources:
> > >
> > > $$
> > > \\min_\\Theta \\sum_i \\pi_i R_{D_i}(\\Theta),
> > > $$
> > >
> > > which encourages the model to fit the *mixture* of source domains. This often leads to features that are highly source-specific and do not transfer to unseen domains.
> > >
> > > In contrast, our episodic meta-learning framework treats each held-out domain $D_i$ as a **separate task**:
> > >
> > > * **inner-loop:** learn from $D_{-i}$
> > > * **outer-loop:** evaluate on $D_i$ as a *virtual unseen target*
> > > * **repeat** for all $i$
> > >
> > > This structure forces the model to succeed across *multiple simulated unseen-domain tasks*, rather than only the training mixture. Any domain-specific shortcut that works only for $D_{-i}$ but fails on $D_i$ is penalized in the meta-objective:
> > >
> > > $$
> > > \\min_\\Theta E_{i} [ R_{D_i}(\\Theta'_{-i}) ]
> > > $$
> > >
> > > As a result, the model is driven toward **domain-invariant, transferable representations**, which aligns with the shift-sensitive terms in our theoretical bound.
> > >
> > > > **In summary, ERM learns features that fit the source mixture, whereas meta-learning learns features that generalize across virtual unseen domains — the core requirement of domain generalization.**

---

### Official Review · Reviewer_Eovd · 2025-10-31

**Soundness:** 3
**Presentation:** 3
**Contribution:** 3
**Rating:** 6
**Confidence:** 4

**Summary:**

This paper addresses the challenging problem of domain generalisation under compound distribution shifts, where both the marginal label distribution and the conditional feature distribution differ across domains. The authors first derive a new theoretical upper bound, then prove that the practical Domain-Class Distribution Alignment loss can upper-bound the Wasserstein distance term in this bound, enabling a tractable optimisation objective. They propose RC-Align, a meta-learning framework that minimises this composite risk bound through a combination of cross-entropy and DA losses within a leave-one-domain-out training protocol.

**Strengths:**

The paper introduces a principled and interpretable risk decomposition that explicitly separates the effects of prior and feature distribution shifts. Empirically verify the correlation between DA loss and generalisation gap, and conduct ablations to show the complementary effects of the DA loss, meta-learning, and Manifold Mixup.

**Weaknesses:**

1. The generalization under concurrent marginal and conditional distribution shifts has been extensively studied in prior works (e.g., Hu et al., 2020; Tan et al., 2024), suggesting that this might not be a critical gap. However, this does not detract from the systemic and insightful theoretical framework presented by the authors.

2. A primary theoretical concern is that the PL condition represents a strong assumption regarding the non-convex loss landscapes inherent to deep neural networks (as acknowledged by the authors in the Appendix). This analysis should therefore be perceived as offering descriptive insights into the algorithm's dynamics within an idealised context, rather than providing stringent guarantees.

3. The DA loss is optimised by comparing feature distances between similar and dissimilar classes, whereas the Wasserstein distance evaluates differences in global distributions. The introduction of intermediate quantities and constants in the proof could render the bound relatively loose. This may reduce the theoretical robustness of the assertion that minimising the DA loss directly equates to minimising feature distribution mismatch.

4.  The analysis presented in Fig. 1 shows a strong correlation between DA loss and the generalisation gap. However, additional ablation studies isolating DA loss’s causal impact on robustness would further substantiate the claim, as potential confounding variables might be present.

5. Table 2 does not include the DomainNet results as presented in Table 3, leading to potential inconsistency in the results representation. Including these results would enhance the comprehensiveness of the analysis.

**Questions:**

1.While the generalization under concurrent marginal and conditional distribution shifts has been explored in previous works, emphasize how your theoretical framework offers a unique perspective or addresses gaps that these studies have not fully tackled.

2. Discuss the implications of the PL condition as a strong assumption in your analysis. Discuss any potential limitations and how these insights can still advance understanding in the field.

3. In regards to optimizing the DA loss, provide further clarification on the role of intermediate quantities and constants in your proof.

4. In response to the suggestion for additional ablation studies in Fig. 1, propose any current or future experiments that could better isolate the causal impact of DA loss on robustness.

5. Explain the discrepancy in the results between Table 2 and Table 3. If available, provide the missing DomainNet results in Table 2.

---

> ### Author Response · Authors · 2025-11-19
>
> ### Response to Reviewer Eovd
>
> *   **Q1(W1). Compound shifts have been studied in prior works. What unique perspective does your theoretical framework offer?**
>
>     A1. Please refer to '1. Regarding Novelty' in the Common Responses.
>
> *   **Q2(W2). The PL condition is a strong assumption in deep learning. Discuss the implications and limitations.**
>
>     A2. We agree. We acknowledge in the paper (L241-244) and Appendix C.7 that the PL condition is a strong assumption. We used it not for global optimization guarantees, but as a standard analytical tool (Karimi et al., 2016) **to understand how the inner adaptation step of meta-learning operates**. The analysis shows how effectively the model parameters are improved locally, providing the insight that the inner adaptation successfully reduces the alignment objective (Proposition 1). Importantly, our main generalization bounds (Theorems 1-3) do not rely on the PL condition.
>
> *   **Q3(W3). Intermediate quantities and constants could render the bound loose.**
>
>     A3. Please refer to '2. Connection between DA Loss and Wasserstein Distance' in the Common Responses.
>
> *   **Q4(W4). Additional ablation studies isolating DA loss’s causal impact on robustness are needed for Fig. 1.**
>
>     A4. Thank you for the constructive suggestion. We analyzed the isolated impact of the DA loss in Table 3 (Ablation Study). Specifically, comparing **"ERM" vs. "ERM + DA Loss"** (isolating the DA loss effect without meta-learning), the Average Acc improved by 2.1%p (77.5% → 79.6%) and Worst Acc by 2.6%p (64.3% → 66.9%) on VLCS. This provides direct causal evidence for the impact of DA loss on robustness.
>
> *   **Q5(W5). Table 2 does not include the DomainNet results as presented in Table 3, leading to potential inconsistency.**
>     **A5. We clarify this concern by addressing two possible interpretations of the question:**
>
>     ### Interpretation 1 — Comparison between Table 2 and Table 3
>     * **Table 3** reports an ablation study conducted only on the VLCS dataset.
>     * **Table 2 (“MDLT Summary”)** already includes DomainNet-MLT results in the rightmost column (Average: 63.8, Worst: 33.8).
>     * Since the two tables serve different purposes and are based on different datasets, there is no inconsistency between them.
>
>     ### Interpretation 2 — On including DomainNet (Standard) results in Table 1
>     If the concern is that **Table 1 (Standard DG)** should also include DomainNet (Standard) results, the exclusion was a deliberate choice aimed at maintaining fairness across benchmarks.
>
>     Several key baselines do not provide public code, which limits our ability to reproduce their results under the standardized **DomainBed** framework:
>
>     * **MetaReg (Balaji et al., 2018)** does not offer an official implementation and uses an experimental setup that differs notably from DomainBed (Gulrajani & Lopez-Paz, 2021) — for example, a random 90/10 train–validation split, SGD-based training, batch size 64, and dataset-specific hyperparameter sweeps — making a consistent reproduction difficult.
>     * **iDAG (Huang et al., 2023)** likewise provides no official implementation, preventing evaluation under a unified setting.
>     * **Some influential baselines (e.g., MLIR, SAGM, Arith)** report only overall DomainNet accuracy without domain-wise results (clipart, infograph, painting, quickdraw, real, sketch), which are essential for Standard DG evaluation.
>
>     Given these factors, establishing fair and methodologically aligned DomainNet comparisons across a broad set of baselines is difficult within a unified experimental protocol.
>
>     By contrast, the four datasets in Table 1 are widely benchmarked and fully reproducible within DomainBed, enabling more comprehensive and reliable comparison.
>
>     We have additionally organized results comparing our method with the remaining baselines not listed above, and we would be glad to share them if this would be helpful.

---

### Official Review · Reviewer_RSKH · 2025-11-01

**Soundness:** 2
**Presentation:** 2
**Contribution:** 2
**Rating:** 4
**Confidence:** 3

**Summary:**

This paper addresses domain generalization (DG) under compound distribution shifts where both the marginal label distribution P(Y) and conditional distribution P(X|Y) vary across domains. The authors propose RC-Align, a meta-learning framework grounded in a novel theoretical upper bound that explicitly decomposes generalization risk into prior shift and feature shift components. The method uses a Domain-Class Distribution Alignment (DA) loss combined with cross-entropy in a MAML-style meta-learning procedure. Experiments on standard DG benchmarks and Multi-Domain Long-Tailed Recognition (MDLT) settings demonstrate state-of-the-art performance.

**Strengths:**

1. The paper provides a clean theoretical decomposition of domain generalization risk into interpretable components (prior shift and feature shift).
2. Good performance on standard DG and MDLT benchmarks.

**Weaknesses:**

1. While the decomposition is useful, the individual components (domain alignment, meta-learning for DG) are well-established. The main contribution is combining them with theoretical justification, but the theoretical tools (Wasserstein distance bounds, InfoNCE decomposition) are standard.

2. The definition of $\pi$ in Theorem 1 is missing.

3. Although the theory motivates minimizing Wasserstein feature distance, the implemented DA loss is a heuristic contrastive loss that aligns features with class centroids. The connection between this loss and the Wasserstein bound is qualitative, not quantitative. The actual training objective may not truly minimize the theoretical upper bound.

**Questions:**

See Weaknesses.

---

> ### Author Response · Authors · 2025-11-19
>
> ### Response to Reviewer RSKH
>
> * **W1. Individual components and theoretical tools are well-established.**
>
>     A1. Please refer to '1. Regarding Novelty' in the Common Responses. Our key contribution is providing novel theoretical insights and a justified methodology for compound distribution shifts using these tools.
>
> * **W2. The definition of $\pi$ in Theorem 1 is missing.**
>
>     A2. Thank you for the question. We apologize if the definition was not immediately apparent. $\pi_{i}(c)$ (and $\pi_{-i}(c)$ for the support mixture) is clearly defined in Section 3 (Line 155) as the marginal label distribution (class prior) of domain $\mathcal{D}_i$ ($P(Y=c|\mathcal{D}_i) := \pi_i(c)$). We suspect the reviewer might have overlooked this definition. In the final version, we will reiterate this definition near Theorem 1 to improve readability.
>
> * **W3. The connection between theory (Wasserstein distance) and implementation (DA loss) is qualitative.**
>
>     A3. Please refer to '2. Connection between DA Loss and Wasserstein Distance' in the Common Responses. Theorem 2 establishes a quantitative, not qualitative, connection.

---

### Author Response · Authors · 2025-11-19

We sincerely thank all the reviewers for their thoughtful reviews and valuable feedback on our manuscript. We are pleased that the reviewers recognized the strengths of our work, including the clean, interpretable, and insightful theoretical decomposition (RSKH, Eovd, Fem2), and the strong performance on both standard DG and challenging MDLT benchmarks (RSKH, Fem2).

The constructive criticisms have been invaluable in improving our work. We have carefully addressed the main concerns raised, such as novelty, the connection between theory and implementation, limitations of theoretical assumptions, and computational cost. We provide detailed responses below.

## Common Responses

### 1. Regarding Novelty (RSKH W1, Fem2 W3, Eovd Q1)

We acknowledge that individual components like **meta-learning** and **domain alignment** exist in the DG literature (e.g., [1–4]). However, the core novelty of our work lies not in a mere combination, but in **proposing the first unified theoretical framework specifically designed for handling compound distribution shifts** (i.e., simultaneous shifts in both the marginal label distribution ($P(Y)$) and the conditional distribution ($P(X \mid Y)$)).

While prior works (e.g., [5–8]) also address such compound or joint shifts of ($P(Y)$ and $P(X \mid Y)$), they typically do so either in application-specific settings (e.g., battery degradation forecasting) or within a domain adaptation framework. Our framework offers a unique perspective in the **domain generalization** setting:

* **Explicit Decomposition.** Theorem 1 explicitly decomposes the DG risk into distinct **Prior Shift** and **Feature Shift** components, in the spirit of earlier DG analyses that relate target risk to source risk plus distribution discrepancy ([9–11]).

* **Quantitative Connection.** Crucially, Theorem 2 proves that the practical Domain–Class Alignment (DA) loss **quantitatively** upper-bounds the Feature Shift term (1-Wasserstein distance), rather than treating alignment as a purely heuristic objective.

This provides a strong, risk-aware theoretical justification for our approach (RC-Align), moving beyond prior works that typically align global features or moments ([1,3,9]) without tying those losses to a principled decomposition of the target risk under joint marginal and conditional shifts.

### 2. Connection between DA Loss and Wasserstein Distance (RSKH W3, Eovd W3/Q3)

Reviewer RSKH noted that the connection between the DA loss and the Wasserstein bound seemed qualitative. However, **Theorem 2 establishes a clear quantitative relationship** between them:

$$ \sum_{c} \pi_{i}(c) W_{1}(P_{i,c}, P_{-i,c}) \leq \mathbb{E}\mathcal{L}\_{\text{DA}}^{(i)}(\theta) \cdot C\_{\text{scat}}^{(i)} + R $$

As Eovd pointed out, the constants ($R$, $C_{\mathrm{scat}}^{(i)}$) might loosen the bound. Nevertheless, the proof guarantees that **minimizing the DA loss directly reduces the theoretical upper bound on class-conditional feature mismatch**, in line with previous work that advocates Wasserstein-based distances as theoretically attractive discrepancy measures ([3,12,13]).

Furthermore, the $C_{\mathrm{scat}}^{(i)}$ (scatter term) is implicitly controlled during $\mathcal{L}_{\mathrm{CE}}$ optimization, as optimizing for classification accuracy inherently promotes intra-class compactness in the feature space. The strong correlation ((r>0.7)) shown in Fig. 1 empirically supports that the DA loss is an effective proxy for reducing the generalization gap.

We also emphasize that, unlike prior work which **uses** Wasserstein distance or OT as a stand-alone alignment loss ([3,12,13]), our Theorem 2 **derives** a *class-conditional* alignment loss as a controllable **surrogate for the Wasserstein term appearing in the risk bound** under compound shift.

---

> ### Author Response · Authors · 2025-11-19
>
> ### 3. Theoretical Limitations and Guarantees (Fem2 W1)
>
> Reviewer Fem2 raised important questions regarding the scope of our theoretical guarantees, specifically concerning the efficacy of ERM for interpolation, the novelty of feature alignment bounds, and generalization to extrapolated domains.
>
> ### R1. "ERM resolves the case when the target domain lies in the convex hull of the source domains." [14]
> We respectfully contest this claim. Our theoretical analysis explicitly demonstrates that ERM does not control the dominant sources of error under compound (marginal + conditional) shifts, even when the target domain lies within the convex hull of the sources [9–11,14].
>
> In Eq. (3), $R\_{D\_{-i}}(\Theta)$ (corresponding to ERM) is is only one component of the upper bound of the target risk. The remaining terms,
>
> $$
> \sum\_{c \in \mathcal{C}} \left| \pi\_i(c) - \pi\_{-i}(c) \right| \cdot R\_{-i,c}(\Theta) \quad \text{and} \quad L\_\ell \cdot \left( \mathbb{E}L^{(i)}\_{\text{DA}} + C^{(i)}\_{\text{scat}} + R \right)
> $$
>
> capture shifts in label priors and class-conditional feature distributions—both entirely uncontrolled by ERM, even under interpolation [7,8].
>
> Thus, minimizing ERM provably fails to guarantee improved target performance unless these shift-sensitive terms are simultaneously reduced [7,8,14].
>
> Empirically, this theoretical insight is strongly supported: ERM underperforms across all compound-shift settings (DG and especially MDLT benchmarks; Tables 1–2). Interpolation alone therefore does not resolve the challenges posed by joint shifts.
>
>
> ### R2. “Bounds based on feature alignment are not new.”
> While alignment concepts have appeared before, our contribution is substantively different and technically stronger.
> We establish a new, explicit connection between a practical Domain-Class Alignment loss and the 1-Wasserstein distance appearing in the risk bound (Theorem 2) [3,12,13].
> This provides a risk-sensitive interpretation of alignment under simultaneous marginal and conditional shift—going beyond prior work [1–4,9,12,13], which typically aligns global features or moments without tying the alignment loss to a principled decomposition of target risk.
> Our bound shows:
> 1. how marginal shift and conditional shift separately contribute to target error, and
> 2. that the proposed DA loss directly upper-bounds the feature-shift term, offering a controllable surrogate for the Wasserstein distance [3,12,13].
>
> Thus, the novelty lies not in “aligning features” but in the risk-aware decomposition and its theoretically justified, optimizable surrogate, which is absent in earlier DG literature [1–4,9–11].
>
>
> ### R3. “The theory does not handle extrapolation.”
> This aligns with the conventional assumptions adopted in DG theory [9–11,14] and therefore reflects a shared theoretical boundary rather than a limitation specific to our framework. Most mixture based DG analyses, both classical and recent, implicitly assume that target domains lie within the span of the source domains. Addressing extrapolated targets would require modeling causal mechanisms or negative transfer effects [15–18], which is orthogonal to the problem setting we study. We will make this scope explicit and highlight extrapolation robustness as an important direction for future work.

---

> ### Author Response · Authors · 2025-11-19
>
> ### References
>
> [1] S. Motiian, M. Piccirilli, D. A. Adjeroh, and G. Doretto, “Unified Deep Supervised Domain Adaptation and Generalization,” in *Proceedings of the IEEE International Conference on Computer Vision (ICCV)*, 2017, pp. 5715–5725.
>
> [2] D. Li, Y. Yang, Y.-Z. Song, and T. M. Hospedales, “Learning to Generalize: Meta-Learning for Domain Generalization,” in *Proceedings of the AAAI Conference on Artificial Intelligence*, 2018.
>
> [3] K. Zhou, Y. Yang, T. Hospedales, and T. Xiang, “Learning to Generate Novel Domains for Domain Generalization,” in *Computer Vision – ECCV 2020*, LNCS 12356, pp. 561–578, 2020.
>
> [4] A. Mehra, B. Kailkhura, P.-Y. Chen, and J. Hamm, “Do Domain Generalization Methods Generalize Well?” in *NeurIPS 2022 Workshop on Machine Learning Safety*, 2022.
>
> [5] R. Tan, X. Lu, M. Cheng, J. Li, J. Huang, and T.-Y. Zhang, “Forecasting Battery Degradation Trajectory under Domain Shift with Domain Generalization,” *Energy Storage Materials*, vol. 72, p. 103725, 2024.
>
> [6] S. Hu, K. Zhang, Z. Chen, and L. Chan, “Domain Generalization via Multidomain Discriminant Analysis,” in *Proceedings of the 35th Conference on Uncertainty in Artificial Intelligence (UAI)*, PMLR 115:292–302, 2020.
>
> [7] X. Liu et al., “Domain Generalization under Conditional and Label Shifts via Variational Bayesian Inference,” in *Proceedings of IJCAI*, 2021.
>
> [8] R. Tachet des Combes, H. Zhao, Y.-X. Wang, and G. Gordon, “Domain Adaptation with Conditional Distribution Matching and Generalized Label Shift,” in *Advances in Neural Information Processing Systems (NeurIPS)*, vol. 33, pp. 1900–1912, 2020.
>
> [9] K. Muandet, D. Balduzzi, and B. Schölkopf, “Domain Generalization via Invariant Feature Representation,” in *Proceedings of the 30th International Conference on Machine Learning (ICML)*, PMLR 28(1):10–18, 2013.
>
> [10] G. Blanchard, A. A. Deshmukh, U. Dogan, G. Lee, and C. Scott, “Domain Generalization by Marginal Transfer Learning,” *Journal of Machine Learning Research*, vol. 22, no. 2, pp. 1–55, 2021.
>
> [11] J. Wang et al., “Generalizing to Unseen Domains: A Survey on Domain Generalization,” *IEEE Transactions on Knowledge and Data Engineering*, vol. 35, no. 8, pp. 8052–8072, 2022.
>
> [12] N. Courty, R. Flamary, D. Tuia, and A. Rakotomamonjy, “Optimal Transport for Domain Adaptation,” *IEEE Transactions on Pattern Analysis and Machine Intelligence*, vol. 39, no. 9, pp. 1853–1865, 2016.
>
> [13] J. Shen, Y. Qu, W. Zhang, and Y. Yu, “Wasserstein Distance Guided Representation Learning for Domain Adaptation,” in *Proceedings of the AAAI Conference on Artificial Intelligence*, vol. 32, no. 1, pp. 4058–4065, 2018.
>
> [14] E. Rosenfeld, P. Ravikumar, and A. Risteski, “An Online Learning Approach to Interpolation and Extrapolation in Domain Generalization,” in *Proceedings of AISTATS*, PMLR 151:2641–2657, 2022.
>
> [15] M. Arjovsky, L. Bottou, I. Gulrajani, and D. Lopez-Paz, “Invariant Risk Minimization,” *arXiv preprint* arXiv:1907.02893, 2019.
>
> [16] D. Mahajan, S. Tople, and A. Sharma, “Domain Generalization using Causal Matching,” in *Proceedings of the 38th International Conference on Machine Learning (ICML)*, PMLR 139:7313–7324, 2021.
>
> [17] P. Sheth, R. Moraffah, K. S. Candan, A. Raglin, and H. Liu, “Domain Generalization – A Causal Perspective,” *arXiv preprint* arXiv:2209.15177, 2022.
>
> [18] F. Lv et al., “Causality Inspired Representation Learning for Domain Generalization,” in *Proceedings of the IEEE/CVF Conference on Computer Vision and Pattern Recognition (CVPR)*, pp. 8036–8046, 2022.

---

### Author Response · Authors · 2025-11-26

Dear Reviewers,

We thank you for the time spent reviewing our work and for your constructive comments. We sincerely hope to have further discussions with you to see if our responses address your concerns.

We have provided detailed explanations in our individual responses to each reviewer.

To summarize, we have:
- Addressed Theoretical & Novelty Concerns (All);
- Strengthened Empirical Evidence & Protocol (Reviewers Eovd, Fem2);
- Clarified Definitions & Connections (Reviewers RSKH, Eovd).

We genuinely hope you will review our responses and consider updating your assessment.
Thank you!

Best Regards,
Authors

---

### Author Response · Authors · 2025-12-03

Dear Reviewers and Area Chair,

We sincerely thank all the reviewers for their engagement during the discussion period. The constructive feedback and insightful questions have been invaluable.

As the discussion phase concludes, we would like to provide a brief wrap-up of the main points addressed during the rebuttal.

### Summary of Recognized Strengths
We are encouraged that the reviewers found our theoretical decomposition of the DG risk under compound shifts to be "clean," "interpretable," "intuitive," and "insightful" (RSKH, Eovd, Fem2). Furthermore, the strong empirical performance of RC-Align on both standard DG and challenging MDLT benchmarks was acknowledged (RSKH, Fem2).

### Key Discussions and Clarifications
The primary discussions revolved following concerns:
- (1) the novelty of our approach,
- (2) the necessity and scope of our theoretical framework, particularly in relation to ERM,
- (3) the connection between our theory and implementation, and
- (4) Empirical Rigor and Protocols.

**1. Clarification of Novelty (RSKH, Fem2, Eovd)**

We clarified that while components like meta-learning and alignment exist, the core novelty of RC-Align lies in proposing the first unified theoretical framework specifically designed for compound distribution shifts in DG. Crucially, Theorem 2 establishes a novel, quantitative connection between the practical DA loss and the Wasserstein distance within our risk decomposition, moving beyond heuristic alignment objectives.

**2. Theoretical Necessity and Limitations vs. ERM (Fem2)**

We engaged in a detailed discussion with Reviewer Fem2 regarding the limitations of ERM and the necessity of our theoretical framework.

**3. ERM Insufficiency and Coupling:**

We clarified that ERM is insufficient under compound shifts. Crucially, the Source Risk and the Distribution Shift terms (Theorem 3) are **coupled** via the learned representation $\Theta$. ERM minimizes Source Risk by learning source-specific shortcuts (e.g., spurious correlations [1, 2, 3]) , which paradoxically **increases** the Shift Term. Therefore, controlling only the risk does not guarantee low target risk.
* **Necessity of Theorems 1 & 2 (Tractable Surrogate):** The Shift Term in Theorem 3 is intractable because the true target distribution $T$ is unknown. Theorems 1 & 2 are essential as they derive the DA loss as a **tractable surrogate** for this intractable shift term.
* **Meta-Learning Objective:** Our meta-learning framework operationalizes this bound by treating each source domain as a virtual target ($D_i$). Unlike ERM, which optimizes for the source mixture, our approach optimizes for generalization across **simulated unseen tasks**. This enables us to jointly control Risk + Shift and drives the model toward domain-invariant, transferable representations.

**3. Connection Between Theory and Implementation (RSKH, Eovd)**

We addressed concerns that the connection between the DA loss and the Wasserstein distance was qualitative or loose. We reiterated that Theorem 2 provides a quantitative upper bound. Minimizing the DA loss directly reduces the theoretical generalization gap, which is strongly supported empirically by the high correlation (r>0.7) shown in Fig. 1.

**4. Empirical Rigor and Protocols (Fem2, Eovd)**

We confirmed that we strictly adhered to the DomainBed standard protocols for hyperparameter tuning (A2 to Fem2). As requested, we also provided a computational cost analysis, showing that RC-Align achieves significant performance gains (+4.1%p over ERM) with a moderate increase in training time (1.45x ERM) (A3 to Fem2).

Our work addresses the challenge of generalization under compound distribution shifts, an area where theoretical guarantees have remained scarce. By establishing a rigorous connection between Wasserstein distance and a tractable optimization objective, we advance the field beyond empirical heuristics toward provably robust representation learning. We believe this framework provides a principled theoretical foundation for handling the complex distribution shifts encountered in real-world applications.

Best Regards,

Authors

**References**

[1] Izmailov, P., Kirichenko, P., Gruver, N., & Wilson, A. Gordon. (2022). On Feature Learning in the Presence of Spurious Correlations. NeurIPS 2022.

[2] Chen, Y., Huang, W., Zhou, K., Bian, Y., Han, B., & Cheng, J. (2023). Understanding and Improving Feature Learning for Out-of-Distribution Generalization. NeurIPS 2023.

[3] Deng, Y., Yang, Y., Mirzasoleiman, B., & Gu, Q. (2023). Robust Learning with Progressive Data Expansion Against Spurious Correlation. NeurIPS 2023.

---

### Meta-Review · Area_Chair_xGWs · 2026-01-05

**Summary:**

The submission proposes a domain generalisation method that makes use of ideas from meta-learning and domain adaptation to optimise a novel DG risk decomposition. The reviewers appreciated the new risk decomposition and promising experimental evaluation results. An issue brought up by more than one reviewer is the fairly small amount of novelty compared to existing approaches, as there are already DG methods that employ quite similar ideas from meta-learning and domain adaptation. The reviewers also raised concerns about the connection between the novel risk decomposition and the proposed algorithm. In particular, from the point of view of developing a tractable training objective, both the informativeness and necessity of the decomposition were questioned.

Connection between theory and practice
Novelty of approach

**Reviewer Concerns:**

Some of the minor concerns about experimental setup (e.g., hyperparameter tuning) were resolved during the discussion. However, the main issues related to novelty and the connection between theory and practice are not adequately resolved.

**Reviewer Scores:**

I believe it is unlikely the reviewers would have changed their scores.

---

### Decision · Program_Chairs · 2026-01-26

Reject